# One substrate many enzymes virtual screening uncovers missing genes of carnitine biosynthesis in human and mouse

Marco Malatesta [1], Emanuele Fornasier[2], Martino Luigi Di Salvo [3], Angela Tramonti [4], Erika Zangelmi[1], Alessio Peracchi [1], Andrea Secchi [1], Eugenia Polverini [5], Gabriele Giachin [2], Roberto Battistutta [2], Roberto Contestabile [3] ✉ & Riccardo Percudani [1] ✉

The increasing availability of experimental and computational protein structures entices their use for function prediction. Here we develop an automated procedure to identify enzymes involved in metabolic reactions by assessing substrate conformations docked to a library of protein structures. By screening AlphaFold-modeled vitamin B6-dependent enzymes, we find that a metric based on catalytically favorable conformations at the enzyme active site performs best (AUROC Score=0.84) in identifying genes associated with known reactions. Applying this procedure, we identify the mammalian gene encoding hydroxytrimethyllysine aldolase (HTMLA), the second enzyme of carnitine biosynthesis. Upon experimental validation, we find that the top-ranked candidates, serine hydroxymethyl transferase (SHMT) 1 and 2, catalyze the HTMLA reaction. However, a mouse protein absent in humans (threonine aldolase; Tha1) catalyzes the reaction more efficiently. Tha1 did not rank highest based on the AlphaFold model, but its rank improved to second place using the experimental crystal structure we determined at 2.26 Å resolution. Our findings suggest that humans have lost a gene involved in carnitine biosynthesis, with HTMLA activity of SHMT partially compensating for its function.

In recent years, the enormous progress in the experimental determination[1,2] and computational prediction[3,4] of protein three-dimensional structures is closing the gap between the 1D and 3D protein information. However, there is still a large gap between structural information and knowledge of protein functions[5,6].

Although the function of proteins is determined by their 3D structure, this information is far less used than the sequence to predict protein function. Homology is the main evidence for protein functional annotation, and the 3D structural information is especially used to extend homology and identify residues important for function[7–11].

Yet, there is a well-established use of protein 3D structures in *molecular docking* screening, in which a database of small molecules (ligands) is screened against a protein (receptor) by assessing binding energy and binding mode. This technique is successfully used for large-scale identification of potential drugs[12,13]. In a complementary approach, a library of receptors is screened against a particular ligand. This *reverse docking* technique is mostly used for finding targets of a known drug[14]. Computational models can be used in these screenings in the absence of experimental structures[15,16].

[1]Department of Chemistry, Life Sciences and Environmental Sustainability, University of Parma, Parma, Italy. [2]Department of Chemical Sciences, University of Padua, Padova, Italy. [3]Istituto Pasteur Italia-Fondazione Cenci Bolognetti and Department of Biochemical Sciences "A. Rossi Fanelli", Sapienza University of Rome, Rome, Italy. [4]Institute of Molecular Biology and Pathology, Italian National Research Council, Rome, Italy. [5]Department of Mathematical, Physical and Computer Sciences, University of Parma, Parma, Italy. ✉e-mail: roberto.contestabile@uniroma1.it; riccardo.percudani@unipr.it

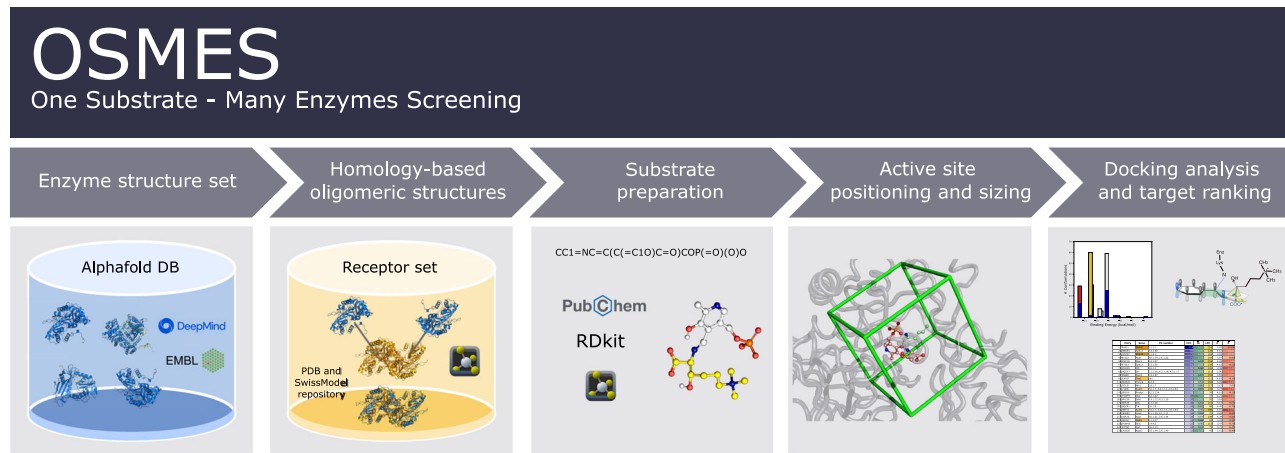

**Fig. 1 | One substrate-many enzymes screening (OSMES) workflow.** Scheme of OSMES. The pipeline consists of 5 main steps performed automatically: (i) Alpha-Fold monomeric models for selected proteins are retrieved; (ii) oligomeric structures are determined with SWISS-MODEL templates; (iii) the substrate is prepared for docking and (iv) used to determine the gridbox size at the active site; (v) finally, the pipeline performs docking analysis and the results are ranked using different methods.

A possible though more challenging use of docking is the matching of enzymes and substrates by predicting the binding of molecules to an enzyme active site[17–19]. Enzymes must bind their substrate molecules with adequate affinity[20,21]. However, binding to an enzyme active site is not sufficient to predict that a molecule would undergo reaction. Since the enzymes have greater affinity for the reaction transition state, docking of molecules mimicking the transition state have been proposed in substrate virtual screening[17]. An alternative strategy is to evaluate whether the binding mode of the docked molecule is suitable for catalysis[19].

Enzymes bind specific substrate conformations that are favorable to the catalyzed reaction. According to the principle of stereoelectronic control, a substrate molecule assumes a conformation at the enzyme active site that minimizes the electronic energy of transition state[21]. A textbook example are the enzymes depending on vitamin B6 (pyridoxal 5′-phosphate; PLP), which catalyze different reactions on amino acids by cleaving different Cα bonds. Cleavage of a particular Cα bond by a specific PLP-dependent enzyme depends on the bond orientation relative to the PLP ring[22–25]. This makes it possible to predict which substrate conformations at the active site favor reactions such as, e.g., racemization, decarboxylation, side-chain cleavage.

Enzymatic reactions for which no genes or proteins are known are present in various metabolic pathways[26,27]. The molecular identification of these 'pathway holes' through a reverse docking approach has now become feasible thanks to the availability of high-quality structures at the proteome level[28].

An example of a metabolic pathway involving a reaction that has not yet been assigned an amino acid sequence is carnitine biosynthesis in mammals[29]. Various eukaryotes synthesize the mitochondrial fatty-acid carrier carnitine through a dedicated four-step pathway. At variance with the fungus *Candida albicans*, in humans and other metazoans the molecular identity of 3-hydroxy-N$^ε$-trimethyllysine aldolase (HTMLA) catalyzing the second step of the pathway is not established, although it is known that the reaction is PLP-dependent[29–31]. This information allows one to restrict the search to a subset of proteins whose full set (PLPome) can be identified by homology[32]. An additional advantage is that the active site of PLP-dependent enzymes is readily identified from the position of the catalytic lysine[33,34].

Here we devise an in silico screening procedure (OSMES: one substrate-many enzymes screening) to identify at the structure level enzymes able to bind a given substrate and catalyze a particular PLP-dependent reaction. First of all, using experimentally known enzyme-substrate combinations, we assess the performance of metrics based on different criteria (binding energy, statistical frequency, catalytically favorable conformation) for the ranking of docked enzyme-substrate complexes. We apply OSMES with the best performing metric to the identification of HTMLA candidates in the human and mouse PLPomes. The results of our screening and subsequent experimental validation allowed us to identify mammalian genes responsible for HTMLA activity in the carnitine biosynthesis pathway.

## Results

### One substrate-many enzymes screening (OSMES) for PLP-dependent enzymes

Here we develop an automated procedure to perform a reverse docking screening of a substrate containing a primary amino group bound to PLP cofactor as a Schiff base (external aldimine; substrate), against a set of 3D enzyme structures of a selected PLPome (enzyme set) (Fig. 1).

As an enzyme set we used PLPomes of *Homo sapiens* and *Mus musculus* retrieved from the B6 database (B6DB; http://bioinformatics. unipr.it/B6db) composed of 56 and 57 genes respectively. For each RefSeq accession number we obtained the corresponding UniProt ID to download the AlphaFold monomer[28] and mark the position of the catalytic lysine useful for subsequent steps. In this first step, we discarded proteins without a conserved catalytic lysine, namely AZIN1, AZIN2, SPTLC1 and PDXDC1 in both sets and Ldc1 in the mouse set, obtaining 105 enzyme targets for our analysis (Supplementary Table 1). The vast majority of our targets have AlphaFold models of very high confidence (pLDDT>90 over 90% of residues) for the overall (>80%) and active site (>95%) residues (Supplementary Fig. 1).

Since most PLP-dependent enzymes belong to fold-type I, which is characterized by obligate dimeric association forming two identical active sites at the interface, oligomerization of the monomeric AlphaFold models is a crucial step of the OSMES pipeline. We therefore exploited models available in the SWISS-MODEL Repository (SMR; https://swissmodel.expasy.org/repository) as templates to assemble AlphaFold monomers into oligomeric structures (Fig. 1, step 2). In our set of enzymes, 96 structures were modeled as oligomers, mostly homomers (79 dimers, 13 tetramers) with the exception of SPTLC2 and SPTLC3, which were modeled as hetero-dimers, both associated with SPTLC1. Once the enzyme set is prepared, the procedure automatically builds the covalent adduct between PLP and a substrate molecule with a given PubChem ID, and creates a 3D coordinate file of the external aldimine for docking screening (Fig. 1, step 3). For each enzyme

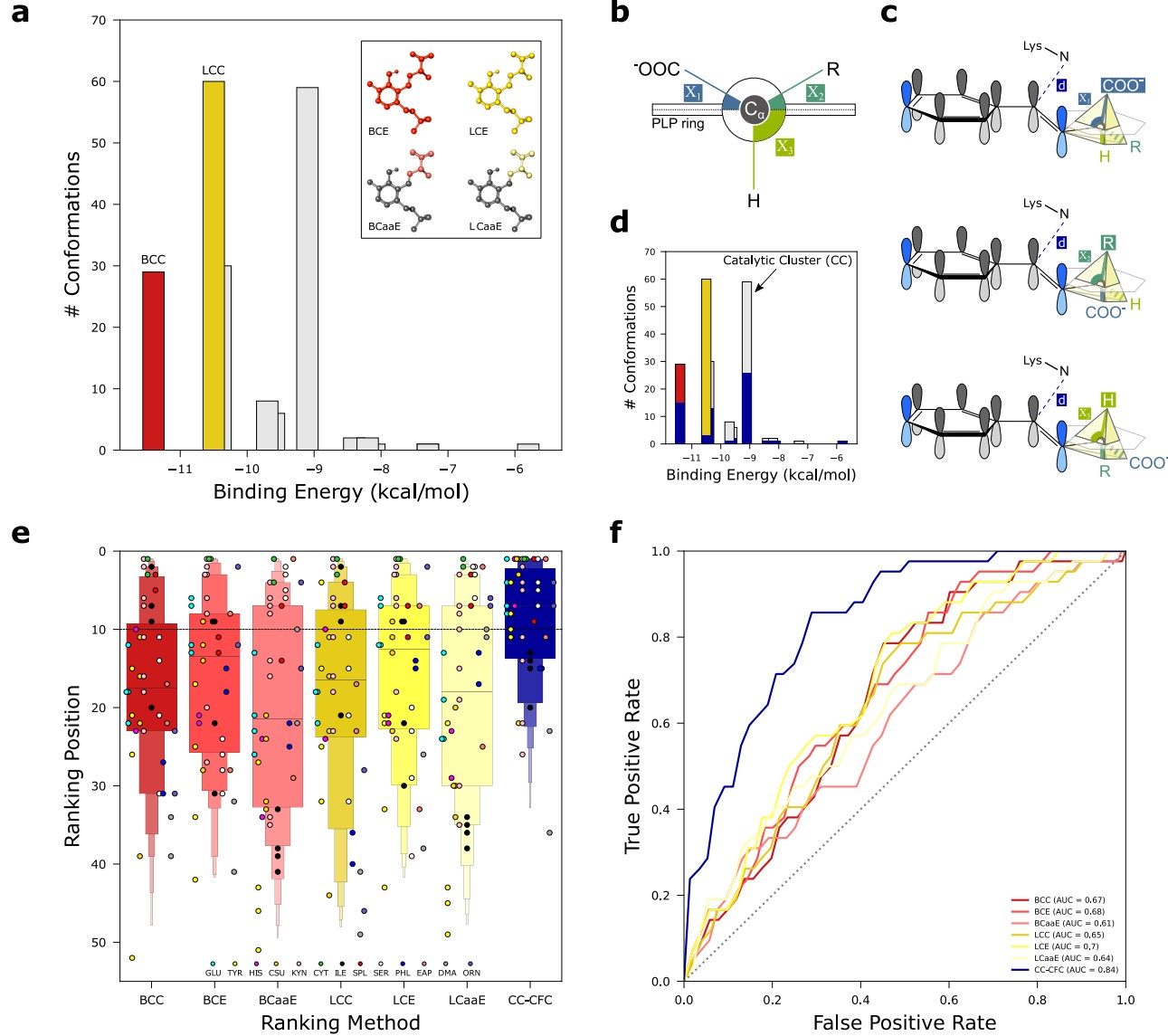

**Fig. 2 | Evaluation of different ranking methods of OSMES with known substrates of PLP-enzymes. a** Representation of the 6 ranking methods related to the best cluster (BC; red tones) and the largest cluster (LC; yellow tones). The bar plot represents the 200 conformations of a single docking run clustered with a 3 Å RMSD threshold; LCC and BCC methods consider the number of conformations in the respective cluster. The atoms of the substrate considered in the energy-based ranking methods (BCE, LCE, BCaaE, LCaaE) are highlighted in the insets. **b** Scheme of the side view of the PLP pyridine ring and the three Cα bonds with the respective angles (χ) with respect to the PLP ring plane. **c** Catalytically favorable conformations (CFC) in the three different PLP-dependent reactions. The conformations from docking analysis are considered CFC if the distance (d) between $N_\varepsilon$ of catalytic lysine and imine carbon is ≤5 Å in the catalytic cluster, and the bond cleaved in the expected reaction (superior circumradius) is nearly orthogonal to the PLP ring (plane), that is its angle χ has the maximum relative value (see Methods). **d** Bar plot

highlighting in blue the number of CFC in different clusters. Black arrow indicates the Catalytic Cluster (CC) which does not always coincide with BC (red) or LC (yellow). **e** Letter-value plot showing the distribution of the validation set ($n = 42$) colored according to the 7 ranking methods. BC related methods are colored in red tones; LC related methods are colored in yellow tones; CC-CFC is colored in blue. Individual dots representing ranking position of positive controls (i.e., enzymes known to act on the substrate) are colored according to substrate (legend); black dashed line delimits the top 10 positions. The band indicates the median, the main box indicates the first and third quartiles with every further minor box splitting the remaining data into two halves. **f** Receiver operating characteristic curve (ROC) for the different ranking methods colored as in panel **e**; the dotted diagonal represents an area under curve (AUROC) value of 0.5. Source data are provided as a Source Data file.

structure, the grid center for docking calculation is positioned at the NZ atom of the catalytic lysine, and the grid size is defined according to the size of the substrate (Fig. 1, step 4) (see Methods).

As a final step (Fig. 1, step 5) the pipeline runs the docking analysis of the substrate against each enzyme structure with AutoDock for Flexible Receptors (ADFR)[35], choosing as flexible residue the same catalytic lysine used to place the grid. The results of the screening are then parsed to rank targets according to different methods (see below).

## Evaluation of catalytically favorable conformations is the best performing metric in OSMES

Before proceeding with OSMES to our case study, we assessed the ability of different ranking methods to identify enzymes involved in particular PLP-dependent reactions. We considered 13 different substrates (Supplementary Fig. 2) against the two PLPomes (human and murine) for a total of 26 screenings evaluated with 7 ranking methods (Fig. 2). In each screening, one or more positive controls represented by enzymes known to catalyze the examined reaction (validation set)

were considered. The validation set consisted of a total of 42 positive controls divided into 14 decarboxylases, 6 aldolases, 14 aminotransferases and 8 other reactions encompassing 4 ammonia-lyases, 2 γ-lyases, and 2 hydrolases (Supplementary Table 2). This set represents about 45% of the 93 human and mouse PLP enzymes with a four-digit EC number.

Among the pose clusters obtained from ADFR analysis, we considered both the lowest-energy cluster (best cluster, BC) and the most populated cluster (largest cluster, LC) (Fig. 2a). The lowest-energy cluster, reflecting the stability of the system, is regarded as the energetically favored one. The most populated cluster, reflecting a higher conformational entropy of the system[36], is regarded as the statistically favored one. For both BC and LC, we ranked the results using three different ranking methods: i) the number of *conformations* in the cluster (BCC and LCC); ii) the lowest binding *energy* of the cluster conformations (BCE and LCE); and, to discount the contribution on the constant moiety of the external aldimine, iii) the lowest binding energy of the cluster conformations without the PLP atoms, considering only the *amino acid* (BCaaE and LCaaE).

In addition to these more canonical criteria, we introduced a ranking method that evaluates the number of catalytically favorable conformations (CFC) based on Dunathan's stereoelectronic hypothesis[22]. According to this widely accepted feature of PLP catalysis, when a compound containing a primary amine group binds covalently to the PLP cofactor to form the external aldimine, the reaction proceeds by breaking the bond more parallel to the π orbitals of the cofactor pyridine ring, or in other words, more orthogonal to the plane formed by the latter. In the case of an α-amino acid, three different cases are possible (Fig. 2b, c), represented by the breaking of the Cα-COOH (as in decarboxylases), Cα-Cβ (as in aldolases), and Cα-Hα bond (as in racemases, aminotransferases, and other lyases). On this basis, for every substrate in our screenings we considered CFC conformations in which the angle ($\chi$) with the PLP ring is maximum for the bond cleaved during the reaction ($\chi_1$ for Cα-COOH; $\chi_2$ for Cα-Cβ; $\chi_3$ for Cα-Hα; Fig. 2b, c). As an additional condition for a CFC, we set an upper threshold of 5 Å for the distance between the NZ atom of catalytic lysine and the imine carbon of external aldimine (Fig. 2c). The cluster with the maximum number of CFC is considered the "catalytic cluster" (CC) and scored by the number of CFC it contains (CC-CFC) (Fig. 2d).

The distribution of the validation test ranked with the 7 different ranking methods shows that with the CC-CFC method the positive controls are generally ranked higher than with other methods (Fig. 2e). Within the CC-CFC distribution, a difference in the performance emerged by categorizing positive controls according to the reaction type, with aminotransferases (A) achieving worse results with respect to other reactions (O) that break the Cα-Hα bond or aldolases (B) and decarboxylases (D) (Supplementary Fig. 3a, b). The good performance obtained by CC-CFC is supported by the area under the receiver operating characteristic (AUROC) that confirms the CC-CFC as the most performing ranking method, with an AUROC = 0.84 compared with 0.7 of LCE, the second best method (Fig. 2f). As an alternative for the second step of the OSMES pipeline (the assembly of oligomeric structures), we considered the use of AlphaFold Multimer[37]. Also in this case, CC-CFC was the best ranking method. However, a slight decrease in the performance was observed with respect to the use of oligomers based on SWISS-MODEL templates (AUROC = 082, Supplementary Fig. 4).

## Application of OSMES to the identification of a missing gene in carnitine biosynthesis

Carnitine biosynthesis begins with release of N[6]-trimethyllysine (TML) from the breakdown of post-translationally modified proteins such as histones, calmodulin, cytochrome c, myosin, etc.[38,39], and involves four enzymatic steps (Fig. 3a). Reactions 1 and 4 are catalyzed by two Fe$^{2+}$-dependent dioxygenases: TML dioxygenase (TMLD) and γ-

butyrobetaine dioxygenase (BBD), which are related by homology; reaction 3 is catalyzed by trimethylamino butyraldehyde dehydrogenase (TMABADH); reaction 2, the aldol cleavage of HTML to generate glycine and TMABA, is catalyzed by HTMLA. Although there is evidence that this activity requires PLP[40,41], the molecular identity of HTMLA in mammals and other metazoans is unknown.

The pathway described above is not universally present in eukaryotes. For instance, it lacks in the yeast *Saccharomyces cerevisiae* and the darkling beetle *Tenebrio molitor*, which require an external supply of carnitine for fat metabolism[42,43]. The distribution of the genes encoding TMLD and BBD in eukaryotes (Supplementary Fig. 5), shows that the known pathway for carnitine biosynthesis is especially present in opisthokonts (fungi and metazoa). However, absence of TMLD and/or BBD in several species, particularly in protostomes, suggests multiple pathway losses, a suitable condition for the identification of missing genes by coevolutionary analysis. This analysis, conducted with a sensitive method of gene coevolution[44] in 1,952 eukaryotic genomes[45] did not reveal an obvious HTMLA candidate, although the best signal among PLP-dependent enzymes was found for an orthogroup annotated as threonine aldolase (Supplementary Table 3). Interestingly, a gene belonging to this group has been previously implicated in *Candida albicans* as HTMLA[30]. A gene homologous to threonine aldolase (*Tha1*) is found in several mammals including mice, but not in humans[46] nor in other species capable of synthesizing carnitine (Supplementary Fig. 5).

Since homology and coevolutionary analysis provided inconclusive evidence on the identification of mammalian HTMLA, we decided to use OSMES on the full set of PLP-dependent enzymes of human and mouse to identify candidates on a structural basis. To this end, we modeled the external aldimine PLP-HTML complex assuming free rotations around rotatable bonds (Fig. 3b) and defined the condition for catalytically favorable conformations of the docked substrate (Fig. 3c): a distance of ≤5 Å of the PLP aldehyde carbon of the substrate from the NZ atom of catalytic lysine and a relative maximum for the $\chi_2$ angle, as expected for the cleavage between Cα-Cβ that occurs in the HTMLA reaction (Fig. 3a).

## HTMLA candidates revealed by OSMES in the human and mouse PLPome

The best performing method (CC-CFC) was used to rank the results of HTML-OSMES against human and murine PLPomes (Fig. 4a). In the two rankings orthologous enzymes are in similar positions, as confirmed by the correlation between the two sets (Spearman $r = 0.83$; Supplementary Fig. 6).

In both rankings, the first hit is the cytosolic serine hydroxymethyltransferase (SHMT1; Shmt1); its mitochondrial version (SHMT2; Shmt2) ranks just after in second (human) and third (mouse) position, as expected from the strong conservation of active sites residues (Supplementary Fig. 7). Interestingly, it has been shown that *E.coli* SHMT can act as an aldolase on β-hydroxylated amino acids, especially with *erythro* configuration[47] that is the configuration adopted by HTML, and it has been proposed that SHMT could be responsible for HTMLA activity in mammals[31]. Descending with the ranking, other potential candidates with tested or predicted aldolase activity and belonging to the same KEGG Reaction Classes as HTMLA (RC00312 and RC00721) are found. These are sphingosine phosphate lyase (SGPL1, Sgpl1; EC: 4.1.2.27), an enzyme anchored to endoplasmic reticulum that catalyzes aldol cleavage forming phosphoethanolamine, and the putative mouse L-threonine aldolase (Tha1), not characterized experimentally but traceable by homology to the yeast low specificity L-threonine aldolase (GLY1, EC: 4.1.2.48). A GLY1 paralog has been genetically characterized as HTMLA in *C. albicans*[30]. Another example of promising candidates is the pair of paralogous enzymes called kynurenine aminotransferases (KYAT1, Kyat1, KYAT3, Kyat3). These enzymes catalyze the transamination of kynurenine into the

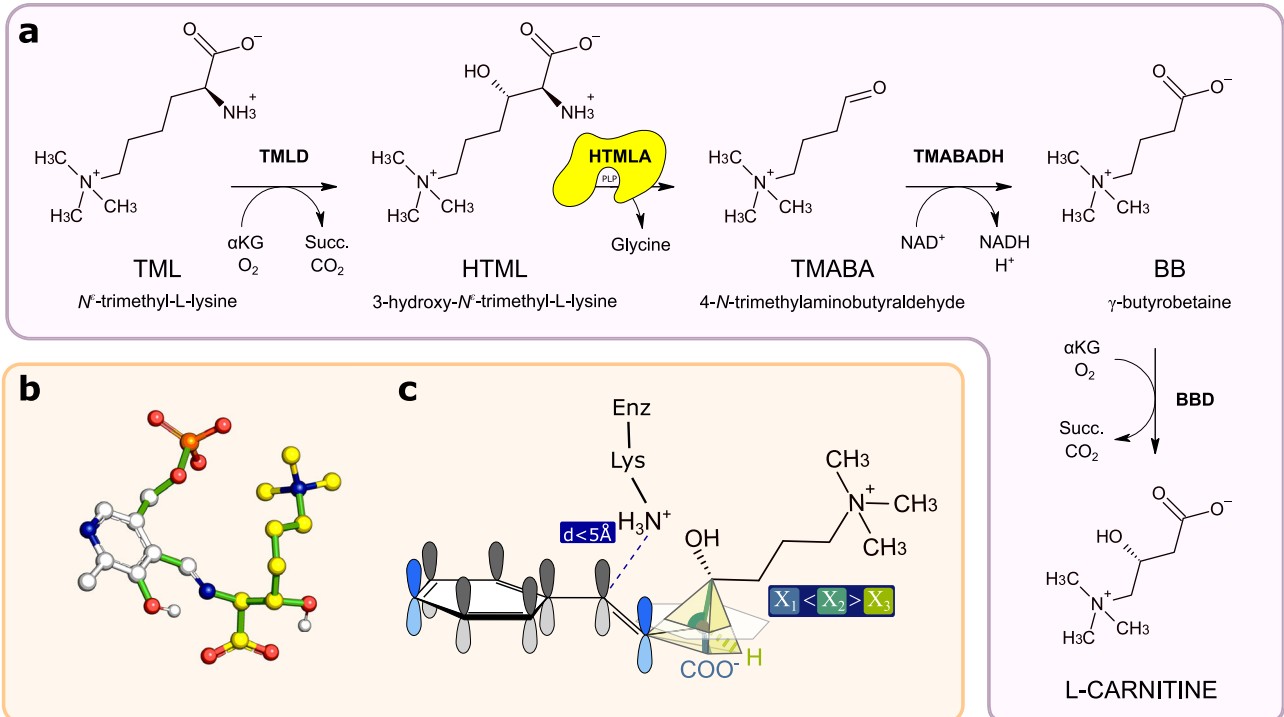

**Fig. 3 | HTMLA, the missing aldolase in animal carnitine biosynthesis.**
**a** Carnitine biosynthetic pathway in animals. HTMLA, the missing enzyme catalyzing the second step of the pathway is highlighted in yellow. **b** Atomic model of the energy-minimized conformation of the HTML-PLP external aldimine used for the OSMES procedure. In yellow the carbon atoms of HTML, in white the carbon atoms of PLP. Non-carbon atoms are colored according to CPK convention. Rotatable bonds are colored in green. **c** Expected geometry of the catalytically favorable conformation of the docked HTML-PLP substrate. The Cα-Cβ bond is considered labile when $\chi_1 < \chi_2 > \chi_3$ and d ≤ 5 Å. Source data are provided as a Source Data file.

corresponding α-keto acid. However, they are also able to catalyze β-lyase reactions toward cysteine-S-conjugate substrates (EC: 4.4.1.13), although the reaction mechanism involves deamination unlike HTMLA[48].

In the catalytic clusters of all the mentioned candidates, ADFR is able to position the PLP cofactor in a binding mode similar to that observed in the available experimental structures of homologous enzymes in complex with PLP (Supplementary Fig. 8). In all four SHMTs and in Tha1, the lowest-energy conformations of HTML-PLP in the catalytic cluster have the Cα-Cβ bond more perpendicular than in the other enzymes (Fig. 4b, Supplementary Fig. 9). By contrast, in the case of both SGPL1 and Sgpl1 (Supplementary Fig. 9), and all KYATs (Fig. 4b, Supplementary Fig. 9), the Cα-COOH ($\chi_3 < \chi_1 > \chi_2$) and Cα-Hα ($\chi_1 < \chi_3 > \chi_2$) bond, respectively, are the most perpendicular and therefore in an unfavorable conformation for aldol cleavage.

In all four KYATs and Tha1, visual inspection of the docked complexes revealed the presence of an aromatic cage (Fig. 4b, Supplementary Fig. 9), characteristic of proteins that bind N-trimethylated substrates, establishing hydrophobic and cation-π interactions with the trimethyl ammonium group[49]. The constant presence of a quaternary amine group in the intermediates of carnitine biosynthesis (Fig. 3a), suggests that an aromatic cage could be a structural feature of all enzymes of the pathway, as evidenced by the BBD structure in complex with γ-butyrobetaine[50], the conservation of the corresponding residues in its homolog TMLD, and the binding mode predicted by docking of the substrate in the TMABADH active site (Supplementary Fig. 10).

### Biochemical validation of HTML-OSMES candidates
For the above reasons, screening candidates KYAT1, SGPL1, SHMT1 and SHMT2 from *Homo sapiens*, and Kyat3 and Tha1 from *Mus musculus*

were chosen for the experimental validation. In addition, we considered screening candidates without previous evidence of aldolase or beta-lyase activity: human ABAT as an example of high-ranking hit, mouse Thnls2 and Oat as mid-ranking hits, and human PSAT1 as a low-ranking hit.

Each protein was produced using optimized conditions in recombinant form to be assayed for HTMLA activity. We obtained soluble expression for all the proteins with the exception of ABAT. Recombinant SHMT1, SHMT2, KYAT1, PSAT1, Kyat3, Thnsl2, Oat were obtained in pure and soluble form after overexpression in *E. coli* (Supplementary Fig. 11a–d; insets). In order to obtain recombinant SGPL1 and Tha1 in the soluble form (Supplementary Fig. 11e,f; insets), they were co-expressed with chaperones (GroEL/GroES) as truncated forms without the N-terminal membrane anchor and mitochondrial signal (Supplementary Fig. 12; see Methods). All the enzymes showed the typical spectrum of protein-bound pyridoxal phosphate in the ketoenamine tautomer, with a peak around 400–430 nm (Supplementary Fig. 11).

Stereospecific (2S,3S) HTML for the activity assays was obtained enzymatically from chemically-synthesized TML (see Methods) by exploiting the first reaction of the pathway (Supplementary Fig. 13). The activity assays show that SHMT1, SHMT2 and Tha1 catalyze the aldol cleavage of HTML; on the contrary, KYAT1, Kyat3, PSAT1, Thnsl2, Oat and SGPL1 are catalytically inactive towards HTML (Fig. 5).

### Human SHMTs catalyze the aldol cleavage of HTML
In the $^1$H NMR spectrum of HTML after addition of SHMT1, the increase of a singlet at 3.55 ppm corresponding to glycine α-protons is visible (Fig. 5a), clearly appearing after 60 min of reaction. TMABA formation is confirmed by 2 distinctive signals at 9.63 ppm and 5.05 ppm of the carbonyl proton and its hydrated form (geminal diol), respectively (Supplementary Fig. 14a).

**a**

### Homo sapiens

| | Entry | Gene | EC number | CC-CFC | $|\sin(\chi_2)|$ | LCC | d | E (kcal/mol) |
|---|---|---|---|---|---|---|---|---|
| 1 | P34896 | SHMT1 | 2.1.2.1 | 116 | 0.81 | 153 | 3.74 | -10.40 |
| 2 | P34897 | SHMT2 | 2.1.2.1 | 102 | 0.80 | 142 | 3.63 | -11.10 |
| 3 | P80404 | ABAT | 2.6.1.19; 2.6.1.22 | 75 | 0.71 | 142 | 3.95 | -9.40 |
| 4 | P20711 | DDC | 4.1.1.28 | 67 | 0.69 | 110 | 3.81 | -8.30 |
| 5 | P32929 | CTH | 4.4.1.1 | 63 | 0.69 | 182 | 3.82 | -11.30 |
| 5 | Q96I15 | SCLY | 4.4.1.16 | 63 | 0.70 | 144 | 3.66 | -9.10 |
| 7 | P13196 | ALAS1 | 2.3.1.37 | 61 | 0.71 | 143 | 3.97 | -10.70 |
| 8 | Q6YP21 | KYAT3 | 2.6.1.7; 4.4.1.13; 2.6.1.63 | 60 | 0.66 | 160 | 3.68 | -12.10 |
| 9 | Q9NUV7 | SPTLC3 | 2.3.1.50 | 57 | 0.74 | 129 | 4.83 | -10.30 |
| 10 | O15270 | SPTLC2 | 2.3.1.50 | 53 | 0.72 | 139 | 4.39 | -10.20 |
| 11 | Q8IUZ5 | PHYKPL | 4.2.3.134 | 46 | 0.72 | 96 | 4.79 | -9.60 |
| 12 | P22557 | ALAS2 | 2.3.1.37 | 45 | 0.74 | 100 | 3.80 | -9.00 |
| 12 | Q9Y697 | NFS1 | 2.8.1.7 | 45 | 0.65 | 89 | 4.24 | -8.90 |
| 14 | Q8N5Z0 | AADAT | 2.6.1.39; 2.6.1.7 | 43 | 0.57 | 128 | 4.18 | -11.00 |
| 15 | Q9GZT4 | SRR | 5.1.1.18; 4.3.1.17; 4.3.1.17 | 42 | 0.69 | 86 | 3.93 | -10.40 |
| 16 | P20132 | SDS | 4.3.1.17; 4.3.1.19 | 41 | 0.66 | 103 | 3.85 | -10.90 |
| 17 | Q86YJ6 | THNSL2 | 4.2.3.- | 39 | 0.51 | 175 | 4.11 | -12.10 |
| 17 | Q16773 | KYAT1 | 2.6.1.7; 4.4.1.13; 2.6.1.64 | 39 | 0.59 | 148 | 4.00 | -11.80 |
| 19 | P21549 | AGXT | 2.6.1.51; 2.6.1.44 | 32 | 0.52 | 141 | 3.97 | -11.80 |
| 19 | P35520 | CBS | 4.2.1.22 | 32 | 0.76 | 55 | 4.13 | -10.80 |
| 21 | P04181 | OAT | 2.6.1.13 | 31 | 0.69 | 69 | 4.93 | -8.60 |
| 21 | P23378 | GLDC | 1.4.4.2 | 31 | 0.55 | 88 | 3.42 | -8.50 |
| 21 | P17174 | GOT1 | 2.6.1.1; 2.6.1.3 | 31 | 0.63 | 82 | 4.03 | -11.90 |
| 24 | O95470 | SGPL1 | 4.1.2.27 | 29 | 0.74 | 93 | 3.30 | -9.90 |
| 25 | Q96GA7 | SDSL | 4.3.1.19; 4.3.1.17 | 28 | 0.73 | 54 | 3.71 | -10.30 |
| 26 | P54687 | BCAT1 | 2.6.1.42 | 26 | 0.57 | 93 | 3.74 | -10.60 |
| 27 | Q9Y600 | CSAD | 4.1.1.29; 4.1.1.11 | 24 | 0.63 | 67 | 3.82 | -10.20 |
| 27 | Q05329 | GAD2 | 4.1.1.15 | 24 | 0.59 | 114 | 3.77 | -10.30 |
| 27 | Q99259 | GAD1 | 4.1.1.15 | 24 | 0.57 | 86 | 3.83 | -10.50 |
| 30 | P11216 | PYGB | 2.4.1.1 | 23 | 0.78 | 42 | 4.01 | -9.30 |
| 31 | P19113 | HDC | 4.1.1.22 | 22 | 0.52 | 100 | 3.75 | -9.50 |
| 31 | Q6ZQY3 | GADL1 | 4.1.1.11; 4.1.1.29 | 22 | 0.56 | 96 | 3.72 | -10.30 |
| 31 | Q96EN8 | MOCOS | 2.8.1.9 | 22 | 0.72 | 75 | 4.86 | -10.60 |
| 34 | P00505 | GOT2 | 2.6.1.1; 2.6.1.7 | 19 | 0.56 | 91 | 3.81 | -12.00 |
| 35 | O15382 | BCAT2 | 2.6.1.42 | 17 | 0.44 | 135 | 3.81 | -11.70 |
| 35 | P17735 | TAT | 2.6.1.5 | 17 | 0.60 | 62 | 4.25 | -9.80 |
| 37 | P06737 | PYGL | 2.4.1.1 | 15 | 0.78 | 131 | 3.65 | -9.70 |
| 38 | Q8TD30 | GPT2 | 2.6.1.2 | 13 | 0.64 | 53 | 3.64 | -11.00 |
| 39 | Q8NHS2 | GOT1L1 | 2.6.1.1 | 11 | 0.49 | 92 | 4.83 | -8.20 |
| 39 | Q9BYV1 | AGXT2 | 2.6.1.44; 2.6.1.40 | 11 | 0.73 | 41 | 3.44 | -8.00 |
| 41 | Q9Y617 | PSAT1 | 2.6.1.52 | 10 | 0.71 | 54 | 4.78 | -9.40 |
| 42 | O94903 | PLPBP | N/A | 9 | 0.41 | 69 | 4.77 | -8.10 |
| 42 | O75600 | GCAT | 2.3.1.29 | 9 | 0.80 | 116 | 3.77 | -9.60 |
| 42 | Q8TBG4 | ETNPPL | 4.2.3.2 | 9 | 0.62 | 54 | 3.45 | -7.30 |
| 42 | Q4AC99 | ACCSL | N/A | 9 | 0.43 | 57 | 3.82 | -10.10 |
| 46 | P11217 | PYGM | 2.4.1.1 | 6 | 0.82 | 137 | 4.18 | -9.70 |
| 46 | Q96QU6 | ACCS | N/A | 6 | 0.34 | 110 | 4.03 | -10.50 |
| 48 | P24298 | GPT | 2.6.1.2 | 3 | 0.40 | 106 | 3.67 | -10.60 |
| 48 | Q8IYQ7 | THNSL1 | N/A | 3 | 0.40 | 117 | 4.18 | -8.90 |
| 50 | Q16719 | KYNU | 3.7.1.3 | 1 | 0.38 | 188 | 3.69 | -9.30 |
| 51 | P11926 | ODC1 | 4.1.1.17 | 0 | 0.38 | 127 | 6.79 | -8.20 |

### Mus musculus

| | Entry | Gene | EC number | CC-CFC | $|\sin(\chi_2)|$ | LCC | d | E (kcal/mol) |
|---|---|---|---|---|---|---|---|---|
| 1 | P50431 | Shmt1 | 2.1.2.1 | 113 | 0.81 | 152 | 3.95 | -10.00 |
| 2 | Q8BG54 | Sptlc3 | 2.3.1.50 | 82 | 0.76 | 136 | 4.61 | -10.10 |
| 3 | Q9CZN7 | Shmt2 | 2.1.2.1 | 77 | 0.80 | 103 | 3.92 | -10.30 |
| 4 | P61922 | Abat | 2.6.1.19; 2.6.1.22 | 71 | 0.79 | 104 | 3.97 | -8.90 |
| 5 | Q8VC19 | Alas1 | 2.3.1.37 | 64 | 0.67 | 155 | 4.29 | -11.40 |
| 6 | P97363 | Sptlc2 | 2.3.1.50 | 61 | 0.70 | 134 | 4.62 | -9.50 |
| 7 | Q8VCN5 | Cth | 4.4.1.1 | 53 | 0.68 | 140 | 3.44 | -12.10 |
| 8 | Q9QZX7 | Srr | 5.1.1.18; 4.3.1.18; 4.3.1.17 | 52 | 0.71 | 111 | 3.90 | -11.00 |
| 9 | Q6XPS7 | Tha1 | N/A | 51 | 0.72 | 111 | 4.83 | -9.10 |
| 9 | P08680 | Alas2 | 2.3.1.37 | 51 | 0.67 | 155 | 3.65 | -10.60 |
| 11 | Q80W22 | Thnsl2 | 4.2.3.- | 46 | 0.53 | 155 | 3.99 | -12.30 |
| 12 | Q9Z1I3 | Nfs1 | 2.8.1.7 | 45 | 0.60 | 117 | 3.67 | -7.60 |
| 12 | Q71RI9 | Kyat3 | 2.6.1.7; 4.4.1.13; 2.6.1.63 | 45 | 0.67 | 135 | 3.77 | -12.10 |
| 14 | Q8R1K4 | Phykpl | 4.2.3.134 | 42 | 0.70 | 76 | 4.61 | -9.10 |
| 15 | Q91WT9 | Cbs | 4.2.1.22 | 38 | 0.84 | 47 | 4.21 | -11.70 |
| 16 | Q8R238 | Sdsl | 4.3.1.17; 4.3.1.19 | 37 | 0.71 | 92 | 3.63 | -10.30 |
| 17 | O88533 | Ddc | 4.1.1.28 | 34 | 0.61 | 110 | 3.57 | -9.00 |
| 18 | Q8QZR1 | Tat | 2.6.1.5 | 28 | 0.62 | 88 | 4.45 | -8.70 |
| 19 | Q8BTY1 | Kyat1 | 2.6.1.7; 4.4.1.13; 2.6.1.64 | 26 | 0.46 | 190 | 4.01 | -13.00 |
| 19 | Q9DBE0 | Csad | 4.1.1.29; 4.1.1.11 | 26 | 0.60 | 70 | 3.85 | -9.80 |
| 21 | O35423 | Agxt | 2.6.1.51; 2.6.1.44 | 25 | 0.47 | 109 | 4.39 | -9.50 |
| 22 | Q8R0X7 | Sgpl1 | 4.1.2.27 | 23 | 0.68 | 83 | 3.34 | -10.40 |
| 23 | P29758 | Oat | 2.6.1.13 | 22 | 0.61 | 75 | 3.71 | -8.40 |
| 23 | Q91W43 | Gldc | 1.4.4.2 | 22 | 0.50 | 113 | 3.47 | -8.70 |
| 23 | Q3UEG6 | Agxt2 | 2.6.1.44; 2.6.1.40 | 22 | 0.75 | 45 | 3.52 | -8.40 |
| 26 | Q8VBT2 | Sds | 4.3.1.17; 4.3.1.19 | 21 | 0.68 | 56 | 3.55 | -10.10 |
| 26 | Q9JLI6 | Scly | 4.4.1.16 | 21 | 0.65 | 76 | 4.10 | -7.80 |
| 28 | P48320 | Gad2 | 4.1.1.15 | 20 | 0.60 | 110 | 3.76 | -10.20 |
| 28 | P05202 | Got2 | 2.6.1.1; 2.6.1.7 | 20 | 0.49 | 80 | 4.17 | -11.30 |
| 30 | Q80WP8 | Gadl1 | 4.1.1.11; 4.1.1.29 | 19 | 0.56 | 106 | 3.68 | -11.10 |
| 31 | O88986 | Gcat | 2.3.1.29 | 18 | 0.43 | 137 | 3.83 | -10.40 |
| 31 | P05201 | Got1 | 2.6.1.1; 2.6.1.3 | 18 | 0.61 | 63 | 4.28 | -10.90 |
| 33 | Q8BH55 | Thnsl1 | N/A | 17 | 0.65 | 97 | 4.58 | -7.50 |
| 34 | O35855 | Bcat2 | 2.6.1.42 | 15 | 0.45 | 96 | 3.78 | -11.90 |
| 34 | Q3UX83 | Accsl | N/A | 15 | 0.77 | 29 | 3.55 | -9.00 |
| 34 | Q8BWU8 | Etnppl | 4.2.3.2 | 15 | 0.76 | 72 | 4.79 | -10.80 |
| 34 | P23738 | Hdc | 4.1.1.22 | 15 | 0.51 | 96 | 3.61 | -9.40 |
| 38 | Q9WVM8 | Aadat | 2.6.1.39; 2.6.1.7 | 14 | 0.33 | 185 | 4.18 | -11.80 |
| 39 | P48318 | Gad1 | 4.1.1.15 | 13 | 0.52 | 95 | 3.66 | -10.50 |
| 40 | Q9ET01 | Pygl | 2.4.1.1 | 12 | 0.74 | 83 | 3.55 | -10.40 |
| 40 | Q8BGT5 | Gpt2 | 2.6.1.2 | 12 | 0.59 | 58 | 4.23 | -10.40 |
| 42 | Q14CH1 | Mocos | 2.8.1.9 | 11 | 0.65 | 85 | 4.92 | -9.00 |
| 42 | Q8CI94 | Pygb | 2.4.1.1 | 11 | 0.80 | 38 | 3.85 | -9.10 |
| 44 | Q9WUB3 | Pygm | 2.4.1.1 | 9 | 0.75 | 108 | 4.02 | -9.50 |
| 45 | A2AIG8 | Accs | N/A | 7 | 0.46 | 69 | 4.00 | -10.20 |
| 45 | P24288 | Bcat1 | 2.6.1.42 | 7 | 0.44 | 47 | 3.72 | -9.50 |
| 47 | Q99K85 | Psat1 | 2.6.1.52 | 5 | 0.54 | 81 | 4.02 | -9.70 |
| 47 | Q7TSV6 | Got1l1 | 2.6.1.1 | 5 | 0.55 | 77 | 4.78 | -7.60 |
| 49 | Q9CXF0 | Kynu | 3.7.1.3 | 2 | 0.47 | 145 | 3.61 | -10.40 |
| 50 | Q8QZR5 | Gpt | 2.6.1.2 | 1 | 0.38 | 66 | 3.95 | -10.50 |
| 50 | Q9Z2Y8 | Plpbp | N/A | 1 | 0.69 | 90 | 4.71 | -5.80 |
| 52 | P00860 | Odc1 | 4.1.1.17 | 0 | 0.52 | 119 | 6.63 | -11.60 |

**b**

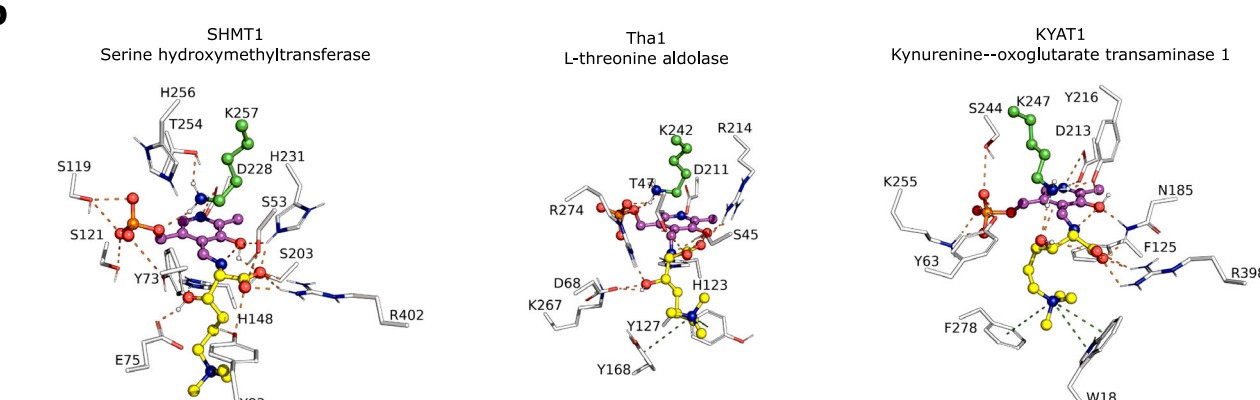

SHMT1
Serine hydroxymethyltransferase

Tha1
L-threonine aldolase

KYAT1
Kynurenine--oxoglutarate transaminase 1

**Fig. 4 | HTMLA candidates identified by HTML-OSMES in human and mouse. a** HTML-OSMES against human and mouse PLPomes ranked with CC-CFC method. Best results (highest for LCC, CC-CFC, $|\sin(\chi_2)|$; lowest for E) in the columns are highlighted with darker colors. $|\sin(\chi_2)|$, d and E columns represent the mean values of CC. In orange are highlighted the enzymes with known β-lyase or aldolase activity. **b** Structural representation of the lowest-energy binding modes among the catalytic clusters obtained by docking of HTML-PLP substrate for SHMT1, Tha1 and KYAT1. Non-carbon atoms are colored according to CPK convention. The conformations are shown with ball-and-sticks and are composed of PLP cofactor (magenta) covalently bound to HTML (yellow), and flexible catalytic lysine (green). The binding site residues (≤4.5 Å from HTML-PLP) are shown in lines labeled with one-letter code and number. Polar interactions between substrate and protein are indicated with orange dashes, while cation-π interactions are indicated with olive dashes. Source data are provided as a Source Data file.

Kinetic characterization of HTML cleavage catalyzed by SHMT1, carried out by a continuous spectrophotometric coupled assay that exploits NAD$^+$ reduction signal at 340 nm in the presence of the third enzyme of the pathway (TMABADH), shows a dependence of the initial velocities on substrate concentrations following Michaelis-Menten kinetics (Fig. 5b). The fitting of data to the Michaelis-Menten equation reveals a catalytic efficiency ($k_{cat}/K_m$) of $32.17 \pm 5.34$ s$^{-1}$ M$^{-1}$ (Supplementary Table 4). We also characterized the enzymatic activity of SHMT2 by spectrophotometric assay (Supplementary Fig. 15i), and measured a lower catalytic efficiency ($6.23 \pm 1.26$ s$^{-1}$ M$^{-1}$) compared to

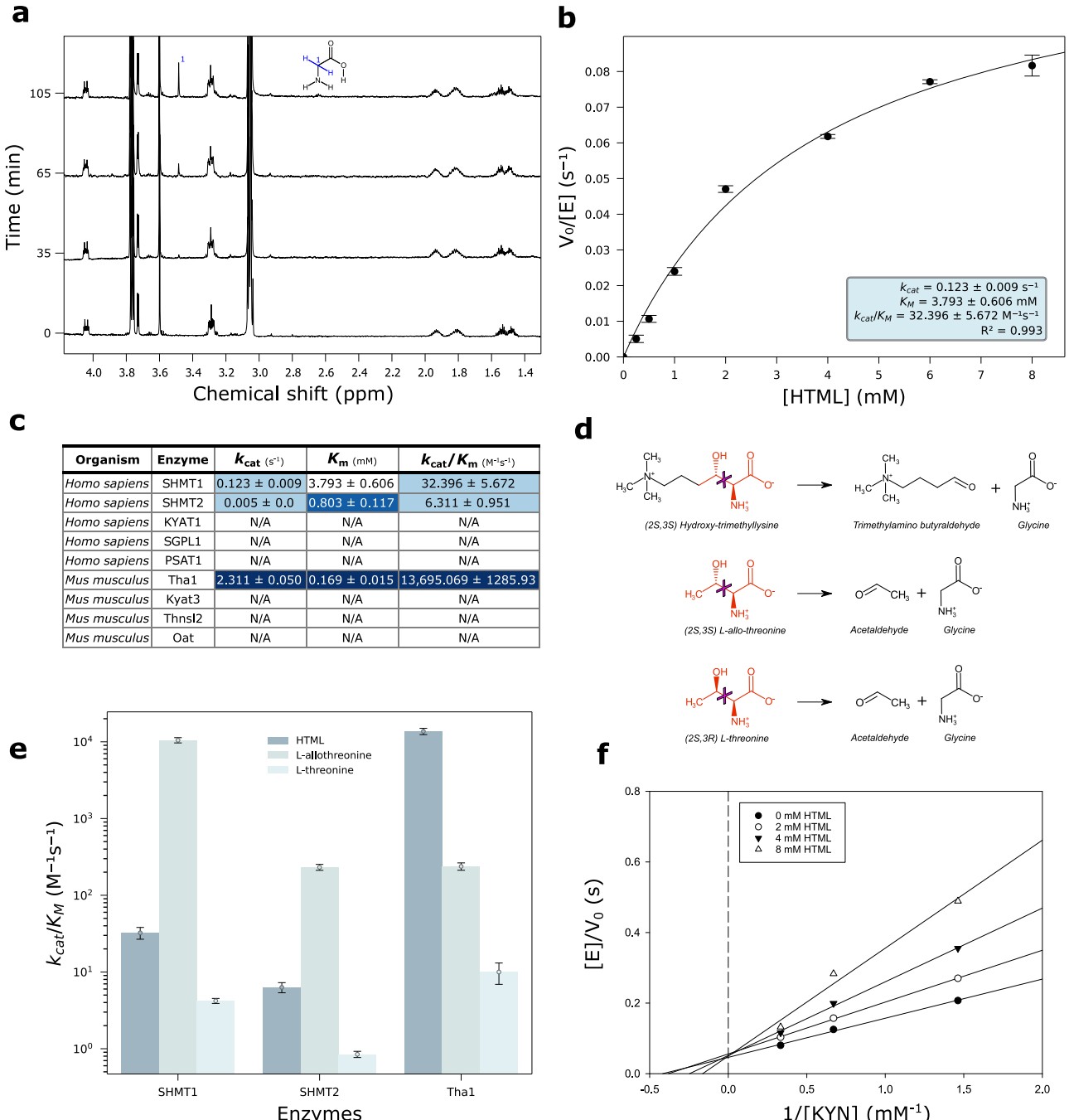

**Fig. 5 | Experimental validation of HTML-OSMES candidates. a** Time-resolved ¹H NMR spectra of SHMT1 activity in the presence of 5 mM HTML at 0, 35, 65 and 105 min. Cα protons singlet of glycine is assigned in the structure. **b** Nonlinear fitting to the Michaelis Menten equation of the dependency on HTML concentrations of the initial reaction velocity of SHMT1 (1 μM). Data are presented as mean values ± SEM; $n = 4$ independent experiments for each point. **c** Kinetic parameters ($k_{cat}$, $K_m$, $k_{cat}/K_m$) of HTMLA reaction of tested enzymes with mean and standard deviation values obtained by nonlinear fitting. The blue-white gradient indicates better (blue) or worse (white) values in each column, where better means higher for $k_{cat}$ and $k_{cat}/K_m$ or lower for $K_m$. **d** Scheme of the broken bond (magenta cross) in the aldol cleavage reactions of HTML, L-*allo*-threonine, L-threonine. In red are the portions common to the three substrates. **e** Bar plot in log scale of the catalytic efficiency of different enzymes (Tha1, SHMT1, SHMT2) with different substrates (HTML, L-*allo*-threonine, L-threonine). The data points correspond to the $k_{cat}/K_m$ values and the error bars represent the standard deviations of the fitting parameters. **f** Lineweaver-Burk double-reciprocal primary plot of the inhibition by HTML of kynurenine aminotransferase activity of KYAT1. The kynurenine concentration ranged from 0.75 to 3 mM. The concentrations of HTML were 0, 2, 4, and 8 mM. Source data are provided as a Source Data file.

SHMT1 (Fig. 5c). In fact, despite a lower $K_m$ (0.80 ± 0.16 mM vs 3.79 ± 0.44 mM) SHMT2 is penalized by a worse $k_{cat}$ (0.005 ± 0.000 s⁻¹ vs 0.122 ± 0.006 s⁻¹). The aldolase activity of human SHMTs towards HTML was not affected by the presence of tetrahydrofolate, a cofactor in the hydroxymethyltransferase reaction catalyzed by the enzyme (Supplementary Fig. 16).

## Mouse threonine aldolase (Tha1) shows higher HTMLA activity than human SHMTs

The ¹H NMR spectrum of HTML after the addition of Tha1, shows peaks with the same chemical shift observed in the reaction with SHMT1, but in higher quantities (Supplementary Fig. 14b), suggesting the same enzymatic activity, but a different efficiency for the two enzymes. A

small upfield shift is visible in the main peak of the trimethylated ammonium protons at 3.11 ppm (Supplementary Fig. 14b).

Kinetic characterization of Tha1 by the same spectrophotometric assay as SHMT1, and fitting to the Michaelis-Menten equation (Supplementary Fig. 15j) resulted in a $k_{cat}$ of $2.311 \pm 0.029\,s^{-1}$ and $K_m$ of $0.169 \pm 0.009\,mM$. Comparison with SHMT1 shows better values for both Tha1 constants and a $k_{cat}/K_m$ ($1.36 \times 10^4\,s^{-1}\,M^{-1}$) about a thousand times greater (Fig. 5c). To test the substrate specificity of Tha1, we evaluated the activity of the enzyme with other β-hydroxylated amino acids: L-threonine and L-*allo*-threonine (Fig. 5d). The enzyme showed activity on both L-threonine and L-*allo*-threonine, but not with the D-enantiomers. However, the preferred substrate of Tha1 is HTML with a catalytic efficiency in the order of $10^4\,s^{-1}\,M^{-1}$, followed by L-*allo*-threonine ($10^2\,s^{-1}\,M^{-1}$) and L-threonine ($10^1\,s^{-1}\,M^{-1}$) (Fig. 5e). These results suggest that Tha1 has a catalytic preference for β-hydroxylated L-amino acids with the *erythro* configuration. With respect to L-*allo*-threonine, the reaction with HTML has a similar $k_{cat}$ but a 50-fold lower $K_m$ (Supplementary Fig. 15j, c), suggesting a higher affinity for the intermediate of the carnitine pathway. The two human SHMTs have a similar a preference for substrates with the *erythro* ($S,S$) configuration, but are much more efficient with L-*allo*-threonine ($\sim 10^4$ for SHMT1, $\sim 10^2$ for SHMT2) than with HTML (Fig. 5e; Supplementary Fig. 15a, b; Supplementary Table 4), which possesses a bulkier side chain (Fig. 5d).

To verify if the preference of Tha1 for the HTML substrate is a feature of threonine aldolase proteins of organisms with the carnitine biosynthesis pathway, we tested the activity of the low-specificity threonine aldolase *e*TA[51] from *E. coli*, which, like other bacteria, does not have carnitine biosynthesis. Recombinant *e*TA was produced in intact form in the homologous host. Characterization of its catalytic efficiency for L-*allo*-threonine and HTML, showed high activity with both substrates with a slight preference for L-*allo*-threonine (Supplementary Fig. 15h, d).

## HTML is a competitive inhibitor of KYAT1
Although KYAT1 is unable to catalyze the aldol cleavage on HTML, the good binding energies obtained with the screening suggest potential binding at the active site. We thus wanted to test if HTML can inhibit KYAT1 activity on L-kynurenine.

In the presence of an α-keto acid, L-kynurenine is converted by KYAT1 to the corresponding keto acid (4-(2-aminophenyl)−2,4-dioxo-butanoate), which rapidly cyclizes to kynurenic acid (Supplementary Fig. 17a). By measuring the spectrophotometric signal at 310 nm of the final product, we were able to observe the progress of the reaction in the absence and in the presence of HTML (Supplementary Fig. 17b, c). After the addition of 0.5 mM of HTML to the reaction mixture, a slowdown of the reaction is observed (Supplementary Fig. 17c), suggesting an inhibitory action. We characterized the initial velocity of kynurenine transamination with increasing concentrations of HTML. The Lineweaver-Burk double reciprocal primary plot shows a family of straight lines intersecting on the y axis, typical of competitive inhibition with a constant $V_{max}$ and an increasing apparent $K_m$ (Fig. 5f). A $K_i$ value of 4 mM was determined by the secondary plot (Supplementary Fig. 17d).

## Crystal structure of mouse Tha1 improves HTML-OSMES results
Although the AlphaFold models in our screening are of high quality overall, there is a disparity in the dataset as evidenced by the different RMSD (root-mean-square deviations) with respect to the templates used for oligomer reconstruction (Supplementary Fig. 18). These differences depend on the availability of experimental structures from the same or closely related species. For instance, in the case of KYAT, SGPL, and SHMT, PDB structures are available from various mammals, including humans[52–55] and mouse[56,57], whereas in the case of Tha1, only PDB structures from distant bacterial homologs are available[51,58]. To verify if the results of our screening for Tha1 are confirmed or

improved with the availability of an experimental structure, we decided to determine the crystal structure of mouse Tha1.

Mouse Tha1 crystallizes in two space groups, in orthorhombic F222 and in monoclinic C2, with one molecule and two molecules in the ASU, respectively. The PLP cofactor is visible only in the monoclinic structure; however, the active site is very similar in the two cases, with only minor differences. The expected tetrameric quaternary structure is formed by crystallographic symmetries, with four identical units in F222 (related by a 222 symmetry) and two identical dimers in C2 (related by a two-fold axis). The RMSD values between the single units (around 0.26–0.28 Å, Supplementary Table 5) indicate that the monoclinic and orthorhombic structures are similar. Also the tetrameric assembly is conserved in the two space groups, with two main interfaces (Fig. 6a). As indicate by data from PISA analysis (Supplementary Table 6), the interface between units A and B (analogous to that between units C and D, termed "main interface") is contributing stronger to the stability of the quaternary structure in comparison with the interface between units A and C (analogous to that between units B and D, termed "secondary interface"). Hence, the tetramer can be considered a dimer (AB + CD) of dimers (A + B and C + D), with the first dissociation being ABCD to AB + CD (as determined by PISA). A comparison with the structure of the *Thermotoga maritima* threonine aldolase (PDB code 1M6S) returned RMSD values between 1.03 and 1.17 Å for the single units (Supplementary Table 5), indicating a significant structure difference even though the secondary structures and the whole quaternary assembly are conserved. The major structural difference is related to an insertion of 10 residues in Tha1 between positions 337–346. In the two enzymes the position of the PLP cofactor is essentially conserved (Fig. 6b). In Tha1, the PLP cofactor, bound to Lys242, is stabilized in the active site by a network of hydrogen bonds and salt bridges with the side chains of Asp211, Arg214 and Thr98 from the same unit, and of Lys267 and Arg274 from the adjacent unit (Fig. 6b). His123 is making an aromatic stacking interaction with the PLP pyridine ring with a relative distance between the rings of 3.7 Å. While the main interface has similar characteristics for the mouse and the *T. maritima* enzymes (as deduced by PISA analysis, see Supplementary Table 6), the secondary interface shows a higher degree of variability. Despite with similar buried area values, around 990–1060 Å², it stronger contributes to the stability of the tetrameric assembly in the *T. maritima* enzyme, with much higher values in ΔG^int (the solvation free energy gain upon formation of the assembly), ΔG^int P-value, and the Complexation Significance Score (CSS). The secondary interface has a more hydrophobic nature in the *T. maritima* enzyme while it is more polar in the mouse enzyme. As a consequence, the tetrameric assembly of the mouse enzyme has a lower stability, with a ΔG^diss (the free energy of assembly dissociation) value of 5–6 kcal/mol compared to the 40.9 kcal/mol of the bacterial enzyme (for the ABCD to AB + CD dissociation). We performed SEC-SAXS experiments that show the presence of a single component with a MW compatible to that of the sum of 4 units, indicating that the mouse enzyme, despite the lower stability, is tetrameric in solution (Supplementary Fig. 19).

We repeated the docking screening by including in the data set the crystallographic structure of Tha1. The HTML-OSMES results show an increase in CFCs compared with what was obtained with the AlphaFold model. In the catalytic cluster, there are 91 CC-CFC within it, compared with 51 in the previous analysis (Fig. 6d, f; Supplementary Table 7). By comparing the two structures, some differences are observed in the side chains of the substrate binding residues (Fig. 6c). There are minor differences in the chain containing the catalytic lysine (e.g. Arg372A), while differences in the position of the residues contributed by the other chains (Tyr168C, Tyr69B) are more pronounced, suggesting that they result mainly from subunit assembly. Most importantly, it is observed that many more conformations of the entire docking analysis with the crystal structure have the relevant bond nearly perpendicular to the plane of the PLP (0°) (Fig. 6e, g), most of

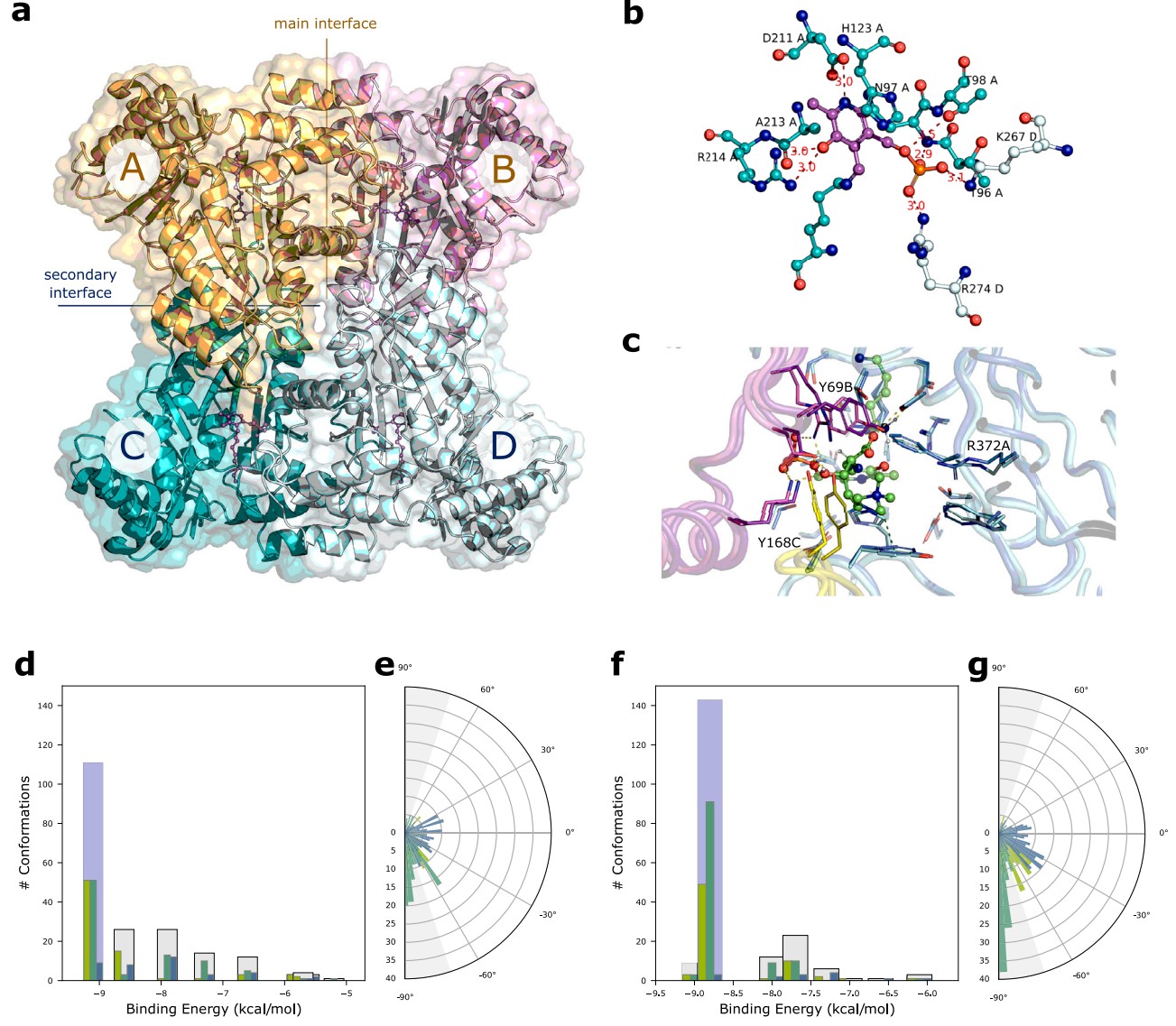

**Fig. 6 | Crystal structure of mouse Tha1 improves HTML-OSMES results.**
**a** Quaternary assembly of Tha1. The four PLP cofactors, one for each unit, are shown in violet ball-and-stick. The main interface, between units A and B (orange/magenta) and the secondary interface between units C and D (teal/pale cyan) are indicated. **b** Active site of Tha1. The main polar interactions of the PLP cofactor (violet) at the interface between subunits A (carbon atoms in teal) and B (carbon atoms in pale cyan) are indicated. Distances in Å. **c** Comparison of AlphaFold model (dark colors) and the Tha1 crystal structure (orthorhombic F222, light colors) docked with HTML-PLP substrate. Different chains are colored in different colors. Non-carbon atoms are colored according to CPK convention. HTML-PLP and flexible catalytic lysine are shown in ball-and-sticks. The binding site residues (≤4.5 Å from HTML-PLP) are shown in sticks. Polar interactions are indicated with orange dashes, cation-π interactions are indicated with olive dashes; residues showing a different position in the model and crystallographic structure are labeled. Clustering of the HTML-PLP conformations at the Tha1 active site obtained with HTML-OSMES applied to AlphaFold model (**d**, **e**) or the crystal structure (**f**, **g**). Bar plots show the distribution of $\chi_1$ (blue), $\chi_2$ (emerald) and $\chi_3$ (kiwi) angles in each cluster, with the catalytic cluster highlighted in light blue. Circular plots show the cumulative distribution of the three $\chi$ angles for all clusters. $|\sin(\chi)| \geq 0.95$ values are defined by gray areas. Source data are provided as a Source Data file.

which have $|\sin(\chi_2)| \geq 0.95$ (gray area). The number of CC-CFC obtained by HTML-OSMES with the experimental structure would have allowed Tha1 to place second in the mouse ranking. Although to a lesser extent, an increase of CC-CFC value was also obtained by the AlphaFold model[59] built with the addition of the Tha1 experimental structure as template (Supplementary Fig. 20; Supplementary Table 7).

**Extension of the OSMES procedure to other enzymes**
To test the possibility of extending the OSMES procedure to a different group of enzymes, we decided to apply OSMES to aldehyde dehydrogenases (EC: 1.2.1.-), a numerous protein family sharing a common catalysis mechanism (Supplementary Fig. 21). A member of aldehyde dehydrogenases, TMABADH, is functionally related to HTMLA, as it catalyzes the subsequent reaction in the biosynthetic pathway (see Fig. 3a). Also in this case we modeled an initial step of the reaction mechanism involving the nucleophilic attack by an active site cysteine to the substrate aldehyde carbon (Supplementary Fig. 21a), with the formation of a covalent intermediate[60]. The subsequent transfer of a hydride ion (H⁻) from this thioester intermediate results in the reduction of NAD(P)+ to NAD(P)H.

We considered substrate orientations in the active site as CFC when the distance between the aldehyde carbon and the catalytic cysteine thiolate was ≤3.5 Å, which is regarded as an upper limit for near attack conformation[60,61]. Using this condition, we applied OSMES to human and murine aldehyde dehydrogenases encompassing two different PFAM domains (Gp_dh_N: PF00044 and Aldedh: PF00171), with a total of 20 and 22 enzymes for human and mouse, respectively (Supplementary Table 10). Since in these proteins the active site is

enclosed within monomeric units, the oligomerization step was not performed.

Six different aldehyde molecules, which are known substrates of eight different enzymes, were used as positive controls for validation (Supplementary Fig. 21b). We observed a performance similar to PLP-dependent enzymes, with CC-CFC as the best ranking method (AUROC score = 0.86) and the energy-based methods (LCE, BCE) as a close second best. Converversely, the ranking methods based on the number of conformations (LCC, BCC) seem to lack discriminative power (Supplementary Fig. 21c, d). In various instances the conformation of the best cluster (BC) corresponded to a catalytically favorable conformation (Supplementary Fig. 21e).

## Discussion

The design of our structure-based screening was motivated by the existence of orphan reactions in biological pathways in which established bioinformatics methods[11,26] fail to identify candidate genes, as in the case of the carnitine biosynthesis pathway investigated here. Since our approach takes into account a single aspect of the catalytic cycle -the formation of a catalytically competent enzyme-substrate complex- it is not anticipated to provide an accurate ranking of enzymatic activities. We have, however, observed that it can aid in the identification of enzymes capable of catalyzing a specific reaction by ranking them in the top positions within a set of proteins. This information can be integrated with previous knowledge and additional bioinformatics evidence (e.g. co-evolution or co-expression with other genes of the pathway), to exclude false positives and identify the most promising candidates for experimental validation.

Our OSMES procedure can be directly applied with modifications of the input parameters to the functional identification of proteins catalyzing a particular set of enzymatic reactions (PLP-dependent reactions and NAD-dependent aldehyde dehydrogenations). PLP-dependent enzymes constitute a variegated subset of biocatalysts present in several metabolic pathways, responsible for more than 300 distinct activities, about 10% of which without an assigned gene (http://www.kegg.jp/kegg/pathway.html; http://bioinformatics.unipr.it/B6db). Aldehyde dehydrogenases have a key role in the detoxification of a large number of reactive aldehydes as well as in the synthesis of biomolecules. The ALDH family has an intricate history of gene duplication and loss[62], complicating homology-based functional assignments. The physiological role and substrate specificity of some human ALDH proteins is still unknown[63,64].

By screening known enzyme-substrate combinations, we observed that a ranking based on catalytically favorable conformations performs best in identifying enzymes responsible for particular reactions. Besides the cases presented here, there are other enzymes in which catalytically productive substrate conformations at the active site can be devised[58-60]. Generally, the determination of catalytically favorable conformations relies on prior knowledge of the catalytic mechanism, which is accessible for a subset of evolutionarily distinct enzymes[65]. The suitability of a CFC method in the screening of enzymatic reactions should thus be assessed on a case by case basis. Interestingly, however, in our validation experiments an acceptable performance was obtained even by scoring methods based on binding energy, which are generally applicable to docking screening.

The application of OSMES to the identification of HTMLA candidates in the mammalian carnitine biosynthesis pathway provides a proof-of-concept of the ability of the screening to predict unknown enzyme-substrate associations on a structural basis. The two top-ranked candidates, SHMT1 and SHMT2, were found to be able to catalyze the HTMLA reaction with a measured catalytic efficiency of ~$10^1$ s$^{-1}$ M$^{-1}$. However, the Tha1 candidate, which was found in the top-10 mouse ranking and is absent in humans, had a HTMLA catalytic efficiency (~1.4 ×$10^4$ s$^{-1}$ M$^{-1}$) about 3 orders of magnitude higher. At

variance with SHMT and other proteins of our set, experimental structures for Tha1 were only available for distant bacterial homologs. Interestingly, the Tha1 ranking in our screening greatly improved by using the crystal structure of the mouse protein or AlphaFold models built taking this information into account.

A surprising result of our experimental validation is that different genes could be responsible for the second step of carnitine biosynthesis in humans and mice. This conclusion, based on bioinformatics and in vitro evidence, is however in line with previous in vivo evidence of greater HTMLA promiscuity than in other reactions of the pathway[31]. In fact, deletion of other genes of the pathway results in the inability of *C. albicans* to grow on fatty acids, whereas deletion of the *gly1* paralog *htmla* only reduces growth on this carbon source, and even the *htmla/gly1* double null strain shows residual growth[30]. Our conclusion that rodents possess a more efficient HTMLA enzyme than humans and other primates in consideration of the low $k_{cat}$ and high $K_m$ of SHMT for the HTML substrate, is supported by the observations that administration of TML to humans results in minimal synthesis of carnitine[66], whereas when TML is given to rats, it is nearly entirely converted into carnitine[67]. Also in line with our results is the observation that the HTMLA activity in human tissues is the lowest amongst the enzymes of the pathway, and is mainly observed in the liver[68], where both *SHMT1* and *SHMT2* are abundantly expressed (https://www.proteinatlas.org/search/shmt).

The *Tha1* phylogeny suggests that this gene has a monophyletic origin in eukaryotes and it has been duplicated only in recent branches of the eukaryotic tree (Supplementary Fig. 22). One of these duplications in Saccharomycetales gave rise to the paralogous gene characterized as *htmla* in *C. albicans*. However, most other fungi possess a single copy of the gene (*gly1*), which is probably responsible for HTMLA activity in fungi. On the other hand, the putative animal ortholog *Tha1* could be responsible for HTMLA activity in those animals in which the gene is present together with the other genes of the pathway. As we found that the mouse enzyme has a strong preference (1000 folds) for HTML towards threonine, the name hydroxytrimethyllysine aldolase (*Htmla*) would be a better descriptor of the mouse gene. It should be noted, however, that orthologous genes are maintained in the budding yeast (YEL046C; *gly1*) and the beetle *Tenebrio molitor* (KAJ3617386) that are known not to produce carnitine[42,43], and in several insect species, such as ants, bees, and wasps, that should lack the biosynthetic pathway as deduced from the absence of TMLD and BBD (see Supplementary Fig. 5). This evidence suggests that Tha1 fulfills additional functional roles, as frequently observed in PLP-dependent enzymes[69,70]. As suggested by the enzyme in vitro activity (see Fig. 5), these additional functions could involve the aldol cleavage of β-hydroxylated L-amino acids with *erythro* configuration.

While *Tha1/Htmla* is present in the majority of eukaryotes, it has been independently lost in various groups of mammals. It is absent in marsupials and some orders of placentals such as Primates and Chiroptera (bats) (see Supplementary Fig. 5). The loss of a functional gene in marsupials is presumably ancient as no trace of the gene can be retrieved in their genome by a tblastn search, whereas is more recent in placentals where pseudogenes are readily identified in several species including humans[46] (Supplementary Fig. 23). The relatively frequent loss of the gene during mammalian evolution can be explained by sufficient supply of carnitine via the biosynthetic pathway ensured by the HTMLA activity of SHMT and/or sufficient exogenous supply of carnitine via the diet. Interestingly, pseudogenization of TMLD is also observed in bats (Supplementary Fig. 23), suggesting loss of carnitine biosynthesis in these species.

According to the subcellular localization of enzymatic activities, carnitine biosynthesis occurs initially in the mitochondria and subsequently in the cytosol. The molecular identity of membrane translocators, responsible for the movements of pathway intermediates

between cellular compartments, and particularly of the postulated mitochondrial TML/HTML antiporter[71], remains unknown.

Carnitine supply is crucial for energy metabolism as it enables the transport of fatty acids into the mitochondria, where they are oxidized to generate ATP. Although not essential to the body's supply of carnitine, nutritional sources are very important in humans, with about 75% of total body carnitine originating from food sources, at least in the presence of an omnivorous diet[72,73]. The results of our study suggest that humans and some other mammals, having lost a gene coding for an enzyme with efficient HTMLA activity, may have a lower output from the biosynthetic pathway and a higher dietary requirement for carnitine.

The experimental characterization of genes and proteins is a severe bottleneck in biology. With high-quality structural models now accessible on a proteome scale, there is a demand for computational methods capable of leveraging this information to enhance our understanding of biological functions. Here we showed that a structure-based screening can guide the identification of proteins catalyzing a particular metabolic reaction and provide evidence for functional assignments independently from sequence-based methods.

## Methods

### Establishment of the human and mouse PLP enzyme set (step 1 and 2)
The PLPome of the considered organisms (*Homo sapiens* and *Mus musculus*) was obtained with the B6DB *whole genome analysis* tool, and each RefSeq accession number was converted to the UniProt one with *OSMES.convert_ac*. The enzyme set was built with AlphaFold models downloaded from https://alphafold.ebi.ac.uk/ in monomeric form and then used for the construction of homo-oligomeric structure with the function *OSMES.build_homo_oligo* that use the *super* function of PyMOL (https://pymol.org/) to structurally superimpose the AF monomers to the best template retrieved by the SWISS-MODEL repository (https://swissmodel.expasy.org/repository).

The criteria to choose the best structure used as alignment template were as follows, in order of priority: database source (first PDB, then SWISS-MODEL), oligomeric state (first the template with the higher number of chain, excluding heteromers), structure resolution (Å or QMEAN). Oligomers for SPTLC2 (AC: P97363, O15270) and SPTLC3 (AC: Q8BG54, Q9NUV7) were built with the function *OSMES.build_oligo_manual* due to their heteromeric association both with the same subunit SPTLC1 (AC: O35704, O15269), obtaining the two heterodimers SPTLC2-SPTLC1 and SPTLC3-SPTLC1. Murine Nfs1 (AC: Q9Z1J3) was built manually with PyMOL with the human template due to the absence of a homodimer in the SWISS-MODEL repository. All the models obtained by the procedure were visually inspected with PyMOL.

For the enzyme set derived from AlphaFold2 Multimer, we employed the UniProt sequence and determined the number of chains based on the SWISS-MODEL templates used in the other set. The *colabfold_batch* command was run on 2 GPU NVIDIA A100 with 80 Gb of RAM.

### Substrate preparation (step 3)
The substrates used for the validation of the procedure were selected to represent amino acids with different properties (negative, positive, hydrophobic, aromatic, etc.). Amino acids were ligated with a covalent bond (imine) between their Nα groups and the aldehydic group of PLP in the external aldimine state. All the substrates used in the reverse docking screening were constructed with an automated process that retrieves the 3D coordinate file of a given PubChem ID (PID) from PubChem (https://pubchem.ncbi.nlm.nih.gov/), binds a user-selected N atom to the PLP provided in SMILE format (CC1=NC=C(C(=C1O)C)COP(=O)([O-])[O-]), adds hydrogens with the *dimorphite_df* program (https://github.com/UnixJunkie/dimorphite_dl) assuming a pH of 7.4,

creates the PDB file and performs energy minimization using the function *MMFFOptimizeMolecule* from *RDKit* (https://www.rdkit.org/) with MMFF94s as forcefield. Once the PDB file is obtained, charges are assigned according to Gasteiger and converted in the pdbqt format (e.g. *HTL_PLP.pdbqt*) with the *prepare_ligand* script of ADFRsuite. An additional txt (e.g. *HTL_PLP.txt*) file is generated that contains the atom ID for the plane, the bonds for the CFC calculation, and the grid box sizes for AGFR obtained during the substrate preparation (see below). The code for steps 1–3 is available in *OSMES.ipynb*.

### Active site positioning and sizing (step 4)
The coordinates for grid positioning are obtained during the enzyme set preparation through i) retrieval of the position of the post translational modification (PTM) lysine in the UniProt database; ii) determination of corresponding the residue number in the PDB file through pairwise alignment (*OSMES.match_fasta_position*) of the UniProt and the PDB sequence converted in FASTA format (*OSMES.pdb2fasta*), and iii) retrieval of the coordinates of the catalytic lysine NZ atom, to be used as the center of the grid. The script output is a tab-separated file (e.g. *Homo_sapiens_coord.tsv*) containing for each pdb file, the coordinates to be used in the AGFR command and required by the OSMES procedure (*OSMES_submit.py*). The box is defined as a cube with a size (*S*) defined during the substrate preparation calculated based on the maximum distance (*maxDist*) among the substrate atoms in the 3D coordinate file, imposing a lower limit of 14 Å.

### Reverse-docking procedure (step 5)
The reverse docking procedure uses the files prepared in the previous steps, i.e. a substrate in pdbqt format (*HTL_PLP.pdbqt*) with the corresponding txt file (*HTL_PLP.txt*) and a dataset of enzyme structures in PDB format with the corresponding coordinates file for the grid box center(*Homo_sapiens_coord.tsv*). These input files are defined in a configuration file (*OSMES.config*) along with other docking parameters. This procedure uses the ADFR suite from AutoDock (https://ccsb.scripps.edu/adfr/), to prepare the pdbqt file of the enzyme structure and to run AGFR and ADFR command for every active site in the set. For each enzyme, all the catalytic lysines of the different chains in the structure were used for docking calculations, e.g. for a tetrameric structure we ran 4 different docking analyses for every chain. For our procedure, we use 200 *nbRuns*, 50,000,000 *maxEvals*, 3 Å for *clusteringRMSDCutoff*, 300 *popSize* and 0.2 Å for *spacing*. The procedure was performed in the SkyLake node (4 INTEL XEON E5-6140 2.3 GHz, 72 cores, and 384 Gb of RAM) of the HPC facility of the University of Parma. A OSMES analysis took about 17 h for each organism considered (~100 active sites).

### Ranking methods
For the classification of the results, 7 different ranking methods were considered: LCC, LCE, LCaaE, BCC, BCE, BCaaE and CC-CFC. LCC and BCC were obtained directly from the summary output file of ADFR command, and correspond to the number of conformations of the largest and best cluster respectively. LCE and BCE were obtained directly from the summary output file of ADFR command, and correspond to energy of the lowest-energy conformation of the largest and best cluster respectively. LCaaE and BCaaE are based on the non binding energies (VdW and electrostatic interactions) of the amino acid-related atoms, excluding those of PLP. These were obtained with *OSMES.calc_ade* that exploits the utility *ade.py* from ADFRsuite and considers only the atoms named with the three-letter code chosen for every substrate in *OSMES.conFigure* To define the CFC in the docking results, we took into account two specific conditions through *OSMES.calc_run*: i) the mean distance of the catalytic cluster between the NZ atom of the catalytic lysine and the imine carbon of external aldimine must be less than 5 Å and; ii) $|\sin(\chi)|$ of the inspected bond should be highest compared to those of the other bonds, and it was

calculated by *OSMES.angle_plane_line* with the following formula:

$$|\sin(\chi)| = \left| \frac{Aa + Bb + Cc}{\sqrt{a^2 + b^2 + c^2}\sqrt{A^2 + B^2 + C^2}} \right|$$

where $Ax + By + Cz = D$ is the plane equation of PLP ring obtained with *OSMES.planeEq* and $ax + by = c$ is the line equation of the inspected bond obtained with *OSMES.lineEq*. $|\sin(\chi)|$ represents the angle between the bond (line) and the PLP ring (plane) and for this reason is [0;1], where 1 is the perfect orthogonality and 0 is the perfect parallelism. For each conformation are calculated 3 different $|\sin(\chi)|$ (i.e. $\chi_1$, $\chi_2$, $\chi_3$) for Cα-COOH, Cα-Cβ and Cα-Hα bonds respectively. All the plots for the analysis of docking results have been obtained with python code available in the *OSMES_result.ipynb* notebook that use Pandas (https://pandas.pydata.org/), Matplotlib (https://matplotlib.org/) and Seaborn (https://seaborn.pydata.org/) libraries.

## OSMES validation with human and mouse aldehyde dehydrogenases
For the OSMES validation with aldehyde dehydrogenases, we considered all the Uniprot records of human and mouse proteome with E.C. 1.2.1.- and a catalytic cysteine in the active site (according to Uniprot features). The position of the catalytic cysteine of ALDH18A1 was determined manually, since the Uniprot record missed this feature. Protein structures were downloaded directly from AFdb and the oligomerization step was skipped since the monomer already has the complete active site. Structures were superimposed to a reference structure depending on the PFAM family (P49189 for Aldedh and P04406 for Gp_dh_N). The grid was placed manually on the 2 reference structures to cover the substrate active site with exclusion of the NAD binding site, using a box of $11 \times 18 \times 11$ Å for Aldedh and $15 \times 16 \times 16$ Å for Gp_dh_N. The docking parameters were the same as PLP-OSMES. The ranking methods considered were LCE, BCE, LCC, BCC and CC-CFC, with the latter corresponding to the number of substrate conformations with a distance ≤3.5 Å between aldehyde carbon of substrate and thiol group of catalytic cysteine. The code is available in the *OSMES_aldh.ipynb* notebook.

## HTML synthesis
HTML was synthesized starting from TML through enzymatic conversion to the hydroxylated form. TML was obtained by chemical synthesis from (2S)−6-amino-2-{[(tert-butoxy)carbonyl]amino}hexanoic acid (purchased from FCH) using a previously described protocol[74].

For the enzymatic HTML synthesis, we recombinantly produced the TMLD enzyme according to a published protocol[75]. We prepared the reaction using triethanolamine instead of the phosphate buffer, which in the presence of $Fe^{2+}$ ion, immediately precipitates. We prepared 100 mL of reaction mixture (α-keto glutarate 15 mM, ascorbate 5 mM, TML 5 mM, $FeSO_4$ 200 μM, TEA 20 mM, DTT 1 mM, TMLD 10 μM) in a flask agitated for 30 min at 37 °C. We finally purified the reaction mixture, that contained the enzyme and the other molecules, with cation exchange chromatography, by exploiting the positively charged N-trimethyl group to isolate HTML from the other negatively charged molecules such as ascorbate, 2-oxoglutarate, and succinate. After reaching pH 5.0 with the addition of HCl, the solution was firstly deprived by the enzyme TMLD through a Vivaspin™ centrifugation, then the flow-through was loaded onto a 50 mL Superloop of ÄKTA pure system FPLC and purified using HiTrap 5 mL SP column. We used 0.2 M HCl to elute the molecule with a gradient of 7 CV. We followed the elution on 210 nm and the fractions of the corresponding peak and flow-through of the column were analyzed by NMR spectra using the setting described below.

## Plasmid construction
SGPL1 (NCBI GeneID: 8879) CDS sequence (XM_006718053.1) was inserted into pET-28b vector. Tha1 (NCBI GeneID: 71776) CDS sequence (NM_027919.4), without the first 40 amino acids corresponding to a predicted mitochondrial signal, was inserted into the pET-28b expression vector. ABAT (NCBI GeneID: 18) CDS sequence (NM_001386611.1) was inserted into the pET-28b-TEV expression vector. All the recombinant plasmids were purchased from GenScript (USA Inc.). The constructs were transformed by electroporation into *E. coli* BL21 with pGRO7 plasmid from Takara™ containing GroEL and GroES chaperons. The authenticity of all constructs was verified by sequence analysis.

## Protein expression and purification
Previously described protocols were used to recombinantly express and purify human SHMT1 and SHMT2[76], human KYAT1 and mouse Kyat3[77], human PSAT1[78], mouse Thnsl2[79] and Oat[80]. Briefly, transformed *E. coli* BL21 Codon Plus cells were grown in a Liter of autoinducing LB broth at 20 °C for 16 h after a pre-induction phase at 37 °C for 8 h, harvested, sonicated and resuspended in suitable buffer (e.g. 50 mM NaH2PO4 pH 7.6, 150 mM NaCl, 20 μM PLP) and purified by affinity chromatography (HisTrap). The *E. coli* clone expressing recombinant rat TMABADH was obtained from Ronald J.A. Wanders (University of Amsterdam). TMABADH was recombinantly expressed and purified as previously described[81]. Tha1, SGPL1 and ABAT expression was performed by inoculating a single colony of every clone in a Liter of autoinducing LB broth obtained by adding 0.5 g/L arabinose, 0.5 g/L glucose and 2 g/L lactose to standard LB medium. Cells were grown at 20 °C for 16 h after a pre-induction phase at 37 °C for 8 h. Cell pellets were resuspended in 50 mL of Lysis Buffer (50 mM $NaH_2PO_4$ pH 7.6, 150 mM NaCl, 20 μM PLP), sonicated (1 s on/off alternatively at 40 W for 30 min) and centrifuged (14,000 × *g* for 40 min at 4 °C). The presence of protein in the soluble fraction was obtained for Tha1 and SGPL1 but not for ABAT, preventing further characterization. Supernatant was loaded onto a 50 mL Superloop of ÄKTA pure system FPLC and purified by Affinity Chromatography (AC) using HisTrap 5 mL FF column. Proteins were washed with Washing Buffer (50 mM $NaH_2PO_4$ pH 7.4, 100 mM KCl, 10% glycerol, 500 mM sucrose, 20 mM $MgCl_2$, 5 mM ATP, 1 mM DTT) to rid of GroEL which would otherwise be found in the elutions (see lane W in Supplementary Fig. 11e); eluted with AC Elution Buffer (20 mM $NaH_2PO_4$ pH 7.6, 150 mM NaCl, 500 mM imidazole). Protein fractions were collected and concentrated by Vivaspin™ centrifugation for dialysis in a Storage Buffer (50 mM $NaH_2PO_4$ pH 7.6, 150 mM NaCl, 1 mM DTT, 5 μM PLP). Tha1 was further purified with a size exclusion chromatography (Supplementary Fig. 19a) using Superdex 200 10/300 Gl column in 50 mM TEA pH 7.6, 150 mM NaCl, 1 mM DTT for crystallization experiments. UV-Vis spectra were collected with JASCO V-750 spectrophotometer. Molar extinction coefficients (ε) for protein quantification were calculated with ProtParam (https://web.expasy.org/protparam/) using the corresponding sequences.

## Aldolase activity assays
The HTMLA activity was measured by coupling the aldol cleavage of HTML with the oxidation of TMABA by NAD⁺ catalyzed by TMABADH. The rate of the reaction was calculated from the rate of appearance in absorbance at 340 nm, due to the formation of NADH, using a value of $\varepsilon_{340} = 6220$ cm⁻¹·M⁻¹. The reaction was carried out using 1 μM SHMT1 or 4 μM SHMT2 in 50 mM potassium phosphate, pH 7.5, containing 1 mM EDTA, 5 mM 2-mercaptoethanol, 0.5 mM NAD⁺ and 8 μM TMABADH, at 37 °C. For Tha1 (1 μM) and *e*TA (0.1 μM) we kept the same conditions but a different temperature (30 °C) due to their low stability at 37 °C. The rate of L-*allo*-threonine and threonine cleavage was measured by coupling the reaction with reduction of the product acetaldehyde by NADH and alcohol dehydrogenase[47]. With L-*allo*-threonine as substrate

were reduced the amount of SHMT enzymes, 0.15 μM for SHMT1 and 1 μM for SHMT2.The rate of the reaction was calculated from the rate of disappearance in absorbance at 340 nm, due to NADH depletion. Initial velocities were collected in a quartz cuvette ($l$ = 1 cm) with JASCO V-750 spectrophotometer and used to calculate kinetic parameters ($k_{cat}$, $K_m$) and their standard deviations (SD) with the *curve_fit* function of the *Scipy* module; SD of $k_{cat}/K_m$ was calculated using error propagation for ratios[82].

### Inhibition characterization of HTML towards human KYAT1

The rate of aminotransferase activity of KYAT1 was measured with a previously described protocol[83]. Briefly, through a continuous spectrophotometric assay at 310 nm ($\Delta\varepsilon = 3625\,M^{-1}\cdot cm^{-1}$) the increase of the signal was monitored due to the higher ε of kynurenic acid compared to that of kynurenine ($4674\,M^{-1}\cdot cm^{-1}$ and $1049\,M^{-1}\cdot cm^{-1}$, respectively). The reaction mixture contains a saturating concentration of α-ketoglutarate as an acceptor of the amino group. Different concentrations of HTML (0, 2, 4, 8 mM) were used for the Lineweaver-Burk double reciprocal primary plot. Initial velocities were collected in a quartz cuvette ($l$ = 0.1 cm) with JASCO V-750 spectrophotometer and plotted with SigmaPlot 14.0.

### NMR spectroscopy

$^1$H NMR spectra were acquired with a JEOL ECZ600R spectrometer in no spinning mode at 25 °C. Samples were loaded in Wilmad ECONOMY NMR tubes, dissolved in 600 μL of $H_2O:D_2O$ (9:1) with simple DANTE presat sequence for $H_2O$ suppression. The reactions were monitored using a DANTE presat array sequence with periods of 300 s for 24 h. NMR experiments were acquired in 50 mM $NaH_2PO_4$ pH 7.0 to avoid signals of organic buffers in $^1$H NMR spectra. NMR spectra were processed and analyzed with MestReNova version 14.2.0 (Mestrelab Research).

### Crystallization and data collection

A frozen aliquot of Tha1 was thawed in ice and concentrated to a final concentration of 3.1 mg/mL and cleared by centrifugation at 17,000 ×*g*. To find the best crystallization conditions crystal screens PACT Premiere HT-96 and Morpheus HT-96 (Molecular Dimensions) were tested mixing 0.2 μL of protein with 0.2 μL of screen against a 40 μL reservoir in MRC 2-lens plates (SWISS-CI) with an Oryx Nano crystallization robot (Douglas instruments). Initial hits appeared in conditions G9 (0.2 M Potassium sodium tartrate tetrahydrate, 0.1 M Bis-Tris propane pH 7.5, 20% w/v PEG 3350), H5 (0.2 M sodium nitrate, 0.1 M Bis-Tris propane pH 8.5, 20% w/v PEG 3350) and H9 (0.2 M Potassium sodium tartrate tetrahydrate, 0.1 M Bis-Tris propane pH 8.5, 20% w/v PEG 3350) of the PACT Premiere HT-96 screen. The protein was finally crystallized using the sitting-drop vapor-diffusion method at 18 °C, mixing 1 μL of Tha1 solution with 1 μL of precipitant solution (0.2 M sodium nitrate, 0.1 M Bis-Tris propane pH 8.5, 20% w/v PEG 3350) and equilibrated against a 80 μL reservoir of precipitant solution in MRC Maxi 48-drops crystallization plates (SWISS-CI). Crystals appeared overnight and finished growing in less than 72 h after the crystallization drops were prepared. For data collections, crystals were fished from the drops and flash-cooled in liquid nitrogen.

### SEC-SAXS measurements and analysis

The Tha1 sample was measured by SEC-SAXS at the ESRF bioSAXS beamline BM29, Grenoble, France. A volume of 250 μL of protein sample at 5.5 mg/mL was loaded on a Superdex 200 10/300 GL column (Cytiva) via a high-performance liquid chromatography device (HPLC, Shimadzu) attached directly to the sample-inlet valve of the BM29 sample changer. The sample was measured in buffer C at 20 °C. The column was equilibrated with at least 3 column volumes to obtain a stable background signal before measurement. All parameters for SAXS analysis are described in Supplementary Table 8 following SAXS

community guidelines[84]. In the SEC-SAXS chromatogram, frames in the region of stable Rg were selected with CHROMIXS and averaged using PRIMUS to yield a single averaged frame per protein sample. Analysis of the overall parameters was carried out by PRIMUS from ATSAS 3.2.1 package[85]. The pair distance distribution functions, P(r), were used to calculate ab initio models with DAMMIF. CRYSOL was used for evaluating and fitting the experimental scattering curve of Tha1 with the corresponding atomic structure solved in this study. Plot and protein model were generated using OriginPro 9.0 and UCSF Chimera software, respectively. SAXS data were deposited into the Small Angle Scattering Biological Data Bank (SASBDB) under accession numbers SASDSU8.

### Structure determination, refinement and analysis

Data collections were performed at ESRF beamline ID23-2. Diffraction data integration and scaling were performed with XDS[86] and the DIALS data-processing package[87], data reduction and analysis with Aimless[88]. Structures were solved by Molecular Replacement with Phaser[89] from Phenix[90], using as search model the structure of L-*allo*-threonine aldolase from *Thermotoga maritima*, PDB code 1M6S. Two crystal forms were identified, in space group F222, with one molecule in the asymmetric unit (ASU), and in space group C2, with two molecules in the ASU. The final refined structures were obtained by alternating cycles of manual refinement with Coot[91] and automatic refinement with phenix.refine[92]. Statistics on data collection and refinement are reported in Supplementary Table 9. A portion of the electron density maps and omit-maps of PLP cofactor are shown in Supplementary Fig. 24. Interface analysis was performed using PISA[93].

### Construction of Tha1 models using crystal structure

Different AlphaFold models are obtained by using the best Tha1 crystal structure (F222) as template using the Colab notebook from ColabFold[59]. By setting *pdb_100* to the parameter *template_mode* with the UniProt sequence of Tha1, the PDB templates used for the prediction were retrieved. The corresponding templates were manually downloaded from PDB and Tha1 crystal structure was added. Then changing to *custom* the *template_mode*, two different models were generated. The first by using as template all the structure retrieved by pdb100 mode plus Tha1 crystal structure (AF_F222) and the second by using as template only the Tha1 crystal structure (AF_only_F222).

### Reporting summary

Further information on research design is available in the Nature Portfolio Reporting Summary linked to this article.

## Data availability

The crystallographic data generated in this study have been deposited in the PDB database under accession codes 8PUS and 8PUM. The SAXS data generated in this study have been deposited in the SASBDB database under accession code SASDSU8. All the other data generated in this study are provided in the Source Data files provided with this paper. Source Data can be accessed from [https://figshare.com/s/65e8b6835e9a76aab0ec].

## Code availability

The datasets and computer code utilized in this research can be accessed via GitHub [https://github.com/lab83bio/OSMES] and have been stored as a permanent archive in Zenodo [https://doi.org/10.5281/zenodo.10709913].

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

## Acknowledgements
We thank Alessandro Paiardini and Michel F. Sanner for discussion, Beatrice Cogliati, Domenico Acquotti, and Davide Cavazzini for technical assistance, Andris Kazaks (Latvian Biomedical Research and Study Centre) for the kind gift of the *E. coli* clone expressing recombinant human TMLD, and Ronald J.A. Wanders (University of Amsterdam) for the kind gift of the *E. coli* clone expressing recombinant rat TMABADH. This work benefited from the equipment and framework of the COMP-HUB and COMP-R initiatives, funded by the "Departments of Excellence" program of the Italian Ministry for University and Research (MIUR, 2018-2022 and MUR, 2023-2027), and from the High Performance Computing facility of the University of Parma, Italy. This work was supported by the Italian Ministry for Education, University and Research PRIN grant 2017483NH8 to R.P. and by Sapienza University of Rome grants RP11715C7CB4FED4, RM11916B51484C08, RM120172A76E4B78 and RM12117A610B653E to R.C. This work was also supported by a grant from Istituto Pasteur Italia – Fondazione Cenci Bolognetti (Research Grant "Anna Tramontano" 2020) to R.C.

## Author contributions
R.P. conceived the study. M.M., R.P., M.L.D.S. and R.C. designed the study. M.M. implemented the OSMES procedure. M.M., M.L.D.S., A.T. and E.Z. conducted protein purification and enzyme assays. E.F., G.G. and R.B. conducted protein crystallization and structural analysis. A.S. performed organic synthesis. A.P. and E.P. contributed critical resources and advice. M.M., R.P. and R.B. wrote the initial draft. All authors contributed to the manuscript.

## Competing interests
The authors declare no competing interests.
