## [Peer Review File · Nature Communications]

One substrate-many enzymes virtual screening uncovers missing genes of carnitine biosynthesis in human and mouseREVIEWER COMMENTS

Reviewer #1 (Remarks to the Author):

This study by Malatesta et al outlines a new automated resource to identify protein-ligand interactions using reverse docking, which is a ligand-centered rather than protein-centered approach. The authors applied this OSMES (one substrate-many enzymes screening) pipeline to a set of PLP-dependent enzymes from human and mouse. After benchmarking with known enzyme-substrate complexes, the authors developed a metric that accounts for catalytically favorable conformations (CFC). Using this criterion, the authors identified a potential “missing” enzyme in the second step of murine carnitine biosynthesis pathway that does not exist in humans. Overall, the manuscript is well written and the methodology appears robust. It is clear this pipeline could be applied to a number of fields, including identifying new protein targets for existing compounds. This reviewer has some specific questions and comments that may be addressed with further explanation in the text.

-the OSMES workflow starts with the AlphaFold monomer and then oligomers are generated with SWISS Model. Have the authors considered using AlphaFold Multimer to generate the oligomers? Have they compared how the performance of the current pipeline compares to using AF Multimer?

-In reference to the PLPome test set, can the authors give some reference point for the scope of the test set? e.g. there were ~100 enzymes and 13 ligands screened. What percentage of the total PLPome does this represent?

Can the authors comment on the difference between the best cluster (lowest energy) and largest cluster (statistically favored) clusters? Why does the BC not overlap with the LC?

-Fig 4a (CC-CFC was used to rank the results of HTML-OSMES against human and murine PLPomes): There are some highly ranked enzymes that do not utilize PLP-HTML as a natural substrate. How would users of this pipeline identify real hits vs false positives?

-Fig 4b and S8: it is very hard to see the labels, particularly those where the text is overlapping

-“Catalytic efficiency” is sometimes referred to as “kinetic efficiency”

-Fig 5c: some boxes are highlighted in different colors but no explanation as to why

-The in vitro activity assays does not necessarily convert to biological specificity. While the authors are careful to use the term “catalytic preference”, a mention of in vivo applicability should be included.

-Table S9: Only 29 or 57 waters were modeled in each structure. Were there specific criteria for why such few atoms were included? Is there some electrochemical/catalytic argument to be made for inclusion/exclusion?

Reviewer #2 (Remarks to the Author):

Malatesta et al. report the identification of the missing enzyme/gene that catalyzes/encodes the second enzyme of carnitine biosynthesis, hydroxytrimethyllysine aldolase (HTMLA). This confirms and unites the snippets of information that have been published since the 1970s until now. Indeed, rodents have a much more efficient conversion of free TML to carnitine than humans. Before this publication this was already attributed to the HTMLA activity, and this study confirms that THA1 is a pseudogene in humans whereas tha1 in mouse/rodents can catalyze the HTMLA reaction, explaining the difference between rodents and man (part of it at least :)). The paper further provides an excellent overview of the carnitine biosynthesis gene distribution in different organisms (which in my view should be brought out of supplementary).

I evaluated the manuscript from a biochemical point of view as I cannot evaluate the structural analysis towards identification of the candidates.

Major point

Page 3/5: My main issue with the paper is that it misses the fifth enzyme of carnitine biosynthesis; the yet to be identified TML/HTML antiporter, see PMID: 17944936, doi:10.1111/j.1742-4658.2007.06108.x. Of course, the message that the only remaining enzyme/gene of carnitine biosynthesis is now identified does not hold, so the fact that there still is one more enzyme/protein missing should be incorporated in the manuscript. In the 17944936 paper TMLD is shown to be present in the mitochondrial matrix and HTML is suggested to be exported to the cytosol. This is relevant as SHMT1 and 2 are cytosolic and mitochondrial enzymes, respectively, and it would support SHMT1 as the most likely HTMLA in humans. This also is in line with the fact that the threonine aldolase activity (in rats at least) is present in the soluble fraction = cytosol of liver homogenates. See PMIDs: 1783176 and 4976434.

I suggest a relatively easy experiment to confirm this: perform a subcellular fractionation experiment in mouse/rat liver using the enzyme assays that are now available in your labs (sup/pellet = cytosol/organelles would likely do the trick). Also, this discussion should be added to the discussion section for a more biochemical / functional accent.

On the side: In this aspect it would be -for later studies- interesting to search for the TML/HTML antiporter. This actually was done in our laboratory but never reported/followed up. We identified Slc25a45 as the TML/HTML antiporter. Free information, do with it what you want.

Minor points

Page 8, third paragraph: the unit for the Km is somewhat cryptic for SHMT1, please add this to the main text to be able to compare Km in mM or uM, that is what enzymologists like. Looking further, I found the Km of SHMT1 in Supplementary table 4 and found it to be 3,8 mM! That is a lot! [now I understand why you did not mention it :)] Especially when compared to tha1 Km and Kcat. The intracellular concentration of TML and HTML are poorly known but this further suggests that the conversion of HTML into TMABA is a slow one in humans, in contrast to that in rodents. This could be included in the discussion.

Page 12, last paragraph: 75% dietary carnitine is for omnivores, not for vegetarians and especially vegans, where more than 75% comes from biosynthesis (and efficient reabsorption in the kidney). Add omnivores and consider adding this nuance.

Supplementary figure 4: very nice figure. Could this be transferred to the main text? It would be a pity if this would be hidden in the supplemental part, it is a great overview of carnitine biosynthesis distribution in nature. One more question; why is *Rattus Norvegicus* not included? Could this be added if data is available as this is one of the best studied organisms when it comes to carnitine biosynthesis.

Supplementary figure 14: please rearrange the panels and cluster them enzyme- or substrate-wise so that it is easier to understand without the legend. Just add the enzyme name in the graph, please.

Reviewer #3 (Remarks to the Author):

This is a manuscript that reports two research stories, one on carnitine biosynthesis in animals that is interesting but more appropriate for a discipline journal, and another on a method to identify function in enzymes, which is innovative and potentially very impactful. Overall, I think the authors should reframe the manuscript to concentrate on the OSMES approach and how it can be deployed to uncover enzyme function in cells.

- 1) On the generality of the OSMES approach. I agree it has been useful for this example, but the authors need to apply it to different systems to convince the community that this is indeed an approach that has merit. This is particularly important for assessing the best performing metric for ranking particular candidates. The metric here is very specific to PLP enzymes—how might it be applied to other enzyme families?
- 2) In the abstract, as stated the outcome seems to be that a gene has been misannotated—yes this may be true (albeit trivial), but is this really the key impact of the work? Isn't the method by which the real gene is uncovered the key impact?
- 3) The finding that the HTML cleavage is less efficient in humans compared to Tha1 (and the conclusion that humans have lost HTML activity) needs to be carefully considered. Yes the kinetic activity in the in vitro assay appears lower, but a variety of factors in the native cell environment could form a perfectly active enzyme—for example, allosteric activation not accounted for in the assay, gene expression levels, pH in the cell, oligomerization. Similarly, in mice Tha1 may be considerably inactive in the native environment. To demonstrate that humans have actually lost this activity, or that it is knocked down, some kind of comparative in vivo assay needs to be done. Metabolomics may be a way forward. Also, it could be that the authors method of finding the real enzyme in humans has failed.
- 4) The authors might consider how protein dynamics play a role in enzyme mechanism and how this knowledge could be reflected in their OSMES approach.
- 5) The Discussion focusses on the carnitine biosynthesis, but this is a less interesting. More consideration should be placed on the OSMES approach and its potential impact.

Minor comments.

- 1) The abstract is perhaps too detailed and not accessible to the general reader. I appreciate the tight word limit, but it isn't immediately obvious to a general reader what has been accomplished.
- 2) Sentence "This allows to predict..." needs editing.
- 3) Are the active site residues formed by both monomers in the dimer? If so, comment on this and

the impact/importance of the 'oligomerization' step in the OSMES workflow.

4) Introduction. The statement that enzymes bind their substrates with high affinity is not true. This would be anticatalytic. Instead, they bind substrates with sufficient affinity. Perhaps the authors are referring to specificity, but even then, many enzymes are not particularly specific. Perhaps rejjig this sentence.

5) The SEC-SAXS data is very nice. The SEC trace appears to be asymmetric – is the oligomeric state of the enzyme changing as a function of protein concentration? If so, what is the likely oligomeric state of the protein in the assay (low [protein]).

Reviewer #4 (Remarks to the Author):

In this work, Malatesta and coworkers develop a computational protocol for identifying enzymes capable of accepting a given substrate and catalyze a particular PLP-dependent reaction. The procedure (OSMES: one substrate-many enzymes screening) is applied to identify 3-hydroxy-N-trimethyllysine aldolase (HTMLA) candidates in human and mice. A few of the identified enzymes are then experimentally tested. In addition to that, a crystal structure is reported for Tha1, which appears to be the most active one. This experimental validation shows that the predicted OSMES ranking is not very accurate, the first two hits for Homo sapiens have very low activity as shown with the kinetic characterization, and similarly for Mus musculus. Although the paper is interesting, in my opinion the protocol is far from being optimal as it does not provide an accurate ranking of the activities, and it is also not clear to me how general the approach is, especially if non-cofactor dependent enzymes are pursued.

I have the following comments:

1. In the OSMES protocol, they use AF2 to generate the 3D structures, but then use SWISS-MODEL oligomerization states for constructing the oligomers. Why not using AF2 multimer from the first place?
2. The protocol constructs a box around the catalytic lysine for docking the PLP-external aldimine. However, many of the X-ray structures of PLP-dependent enzymes are crystallized in the internal aldimine form (Lys covalently bound to PLP), which means that the conformation of the lysine might be different from the one it adopts in the external aldimine intermediate. In addition to that, the external aldimine is optimized using a forcefield instead of DFT (or semi empirics). How good/reliable are the docked external aldimine structures? The comparison of a predicted pose with an X-ray structure with an external aldimine (or similar) would reveal interesting insights in this regard. In Fig. S2 some structures for the validation PLP-external aldimine are shown, but it is hard to assess the quality of these structures. The same happens with the PLP-external aldimine docked in the active site pocket that are shown in figures S7 and S8. The authors should provide more quantitative descriptors (distances, angles, RMSD, etc).
3. They have experimentally tested the most efficient candidates from Homo sapiens and Mus musculus and found that the candidate in mice is the most efficient for the reactivity they are seeking, while the top two candidates from Homo sapiens are less efficient. Based on this, they conclude that humans have lost the gene involved in Carnitine synthesis. Is this stating entirely

accurate? It could be that with the method they have used, they haven't been able to find the best candidate, but a better one to the ones identified could exist.

4. As mentioned before, OSMES ranking is not very accurate, as the most efficient enzyme for mice (called Tha1) is ranked 9th. However, once they obtain the X-ray structure and use it in the OSMES method, they find that Tha1 is ranked second. What differs between the model and X-ray structure? This might suggest that the used models are not of high accuracy.

5. It is not clear to me why the better ranked enzymes in the case of mice were not experimentally characterized. In addition to that, the experimental characterization of enzymes ranked at the low and mid positions in the ranking might also provide relevant information on the quality of the ranking developed.

6. In the validation step, the authors find that within the CC-CFC distribution aminotransferases achieved the worst results as compared to other reactions. Do the authors have an explanation for that?

Minor notes:

1. It is stated that some bonds of the PLP-HTML complex were allowed to rotate (Fig. 3b). However, I could not find the information on the rotatable bonds in Fig. 3b.

We thank the four reviewers for their consideration of our manuscript, valuable inputs, and constructive criticism. Our point-to-point responses to their questions are marked in blue, with the modified parts in the manuscript in bold.

REVIEWER COMMENTS

Reviewer #1 (Remarks to the Author):

This study by Malatesta et al outlines a new automated resource to identify protein-ligand interactions using reverse docking, which is a ligand-centered rather than protein-centered approach. The authors applied this OSMES (one substrate-many enzymes screening) pipeline to a set of PLP-dependent enzymes from human and mouse. After benchmarking with known enzyme-substrate complexes, the authors developed a metric that accounts for catalytically favorable conformations (CFC). Using this criterion, the authors identified a potential “missing” enzyme in the second step of murine carnitine biosynthesis pathway that does not exist in humans. Overall, the manuscript is well written and the methodology appears robust. It is clear this pipeline could be applied to a number of fields, including identifying new protein targets for existing compounds. This reviewer has some specific questions and comments that may be addressed with further explanation in the text.

-the OSMES workflow starts with the AlphaFold monomer and then oligomers are generated with SWISS Model. Have the authors considered using AlphaFold Multimer to generate the oligomers? Have they compared how the performance of the current pipeline compares to using AF Multimer?

When this study was initiated AF Multimer was not available. In the revised manuscript, we analyzed with OSMES the enzyme set for both organisms obtained with AlphaFold Multimer. The results, although similar, are slightly worse than those obtained with superposition of Swiss-Model oligomers (AUROC=0.82 Vs. 0.84). [See also our response to reviewers #4]

Page 5: As an alternative for the second step of the OSMES pipeline (the assembly of oligomeric structures), we considered the use of AlphaFold Multimer³⁶. Also in this case, CC-CFC was the best ranking method. However, a slight decrease in the performance was observed with respect to the use of oligomers based on SWISS-MODEL templates (AUROC=0.82, Supplementary Fig. 4).

-In reference to the PLPome test set, can the authors give some reference point for the scope of the test set? e.g. there were ~100 enzymes and 13 ligands screened. What percentage of the total PLPome does this represent?

As there are 93 PLP-dependent enzymes with a four-digit EC number in the human and mouse set, the 42 enzymes of the evaluation set (corresponding to 13 distinct substrates) represent about 45% of the human and mouse PLP enzymes with known substrates.

Page 5: [The validation set consisted of a total of 42 positive controls...]. This set represents about 45% of the 93 human and mouse PLP enzymes with a four-digit EC number.

Can the authors comment on the difference between the best cluster (lowest energy) and largest cluster (statistically favored) clusters? Why does the BC not overlap with the LC?

If the convergence of a docking simulation is reached (and we have verified that this is the case), the presence of multiple clusters at similar binding energy (within the software error), usually can be ascribed to the possibility of multiple conformations of a ligand inside the binding site, which can reflect the actual disorder present in the complex (Cosconati et al., *Expert Opin Drug Discov.*, 5: 597–607, 2010; Rosenfeld et al., *J. Comput.-Aided Mol. Des.*, 17: 525–536, 2003). This is particularly observable if the number of degrees of freedom during the docking is high. Even if we assume that the best cluster is the one with the lowest energy, however we cannot ignore the cluster with the highest number of elements, that reflects a higher conformational entropy of the system (Chang et al., *Comput Chem*, 29: 1753–1761, 2008) and therefore is statistically favored, that is, there is a higher probability that the system falls down in such a conformational cluster. Therefore a combination of these two metrics can be used to evaluate the results.

Page 5. The lowest-energy cluster, reflecting the stability of the system, is regarded as the energetically favored one. The most populated cluster, reflecting a higher conformational entropy of the system ³⁶, is regarded as the statistically favored one.

-Fig 4a (CC-CFC was used to rank the results of HTML-OSMES against human and murine PLPomes): There are some highly ranked enzymes that do not utilize PLP-HTML as a natural substrate. How would users of this pipeline identify real hits vs false positives?

It is indeed possible to find false positives among highly ranked enzymes. In the revised version we point out that additional evidence from previous knowledge or sequence analysis can be used to identify real hits. In our case we integrated the results of the screening with previous knowledge on the enzyme activities, genetic evidence in yeast, and co-evolutionary information.

Page 11: This information can be integrated with previous knowledge and additional bioinformatics evidence (e.g. co-evolution or co-expression with other genes of the pathway), to exclude false positives and identify the most promising candidates for experimental validation.

-Fig 4b and S8: it is very hard to see the labels, particularly those where the text is overlapping

As suggested, we improved the poorly visible labels by increasing their sizes and limiting them only to residues that interact with the substrate. We also hid the cartoon visualization of the protein for clearer images. The revised figures are **Fig. 4b, 6b** and **Supplementary Fig. 8, 9, 10**.

-“Catalytic efficiency” is sometimes referred to as “kinetic efficiency”
The term “kinetic efficiency” on page 8 has been replaced with “catalytic efficiency” already used elsewhere in the paper.

-Fig 5c: some boxes are highlighted in different colors but no explanation as to why
We have added an explanation in the figure legend of the revised version.

Fig 5c, legend: “The blue-white gradient indicates better (blue) or worse (white) values in each column, where better means higher for k_{cat} and k_{cat}/K_m or lower for K_m .”

-The *in vitro* activity assays does not necessarily convert to biological specificity. While the authors are careful to use the term “catalytic preference”, a mention of *in vivo* applicability should be included.

In the discussion of the revised manuscript, we emphasize that our conclusions on the metabolic pathway **are based on bioinformatics and *in vitro* evidence**, However, we also highlight that they align with *in vivo* evidence accumulated over the past fifty years of research on carnitine metabolism [see also our response to reviewer #3].

-Table S9: Only 29 or 57 waters were modeled in each structure. Were there specific criteria for why such few atoms were included? Is there some electrochemical/catalytic argument to be made for inclusion/exclusion?

As is common practice, water molecules were introduced during refinement (both automatically and manually) on the basis of the electron density and chemical plausibility (mainly because of the possibility to form hydrogen bonds with the protein). The different numbers of water molecules, 29 or 57, reflect the different quality of the electron densities in the two structures, correlated with the maximum resolution, 2.60 and 2.26 Å, respectively. No electrochemical or catalytic arguments were used as criteria for placing water molecules.

Reviewer #2 (Remarks to the Author):

Malatesta et al. report the identification of the missing enzyme/gene that catalyzes/encodes the second enzyme of carnitine biosynthesis, hydroxytrimethyllysine aldolase (HTMLA). This confirms and unites the snippets of information that have been published since the 1970s until now. Indeed, rodents have a much more efficient conversion of free TML to carnitine than humans. Before this publication this was already attributed to the HTMLA activity, and this study confirms that THA1 is a pseudogene in humans whereas *tha1* in mouse/rodents can catalyze the HTMLA reaction, explaining the difference between rodents and man (part of it at least :)). The paper further provides an excellent overview of the carnitine biosynthesis gene distribution in different organisms (which in my view should be brought out of supplementary).

I evaluated the manuscript from a biochemical point of view as I cannot evaluate the structural analysis towards identification of the candidates.

Major point

Page 3/5: My main issue with the paper is that it misses the fifth enzyme of carnitine biosynthesis; the yet to be identified TML/HTML antiporter, see PMID: 17944936, doi:10.1111/j.1742-4658.2007.06108.x. Of course, the message that the only remaining enzyme/gene of carnitine biosynthesis is now identified does not hold, so the fact that there still is one more enzyme/protein missing should be incorporated in the manuscript. In the 17944936 paper TMLD is shown to be present in the mitochondrial matrix and HTML is suggested to be exported to the cytosol. This is relevant as SHMT1 and 2 are cytosolic and mitochondrial enzymes, respectively, and it would support SHMT1 as the most likely HTMLA in humans. This also is in line with the fact that the threonine aldolase activity (in rats at least) is present in the soluble fraction = cytosol of liver homogenates. See PMIDs: 1783176 and 4976434.

I suggest a relatively easy experiment to confirm this: perform a subcellular fractionation experiment in mouse/rat liver using the enzyme assays that are now available in your labs (sup/pellet = cytosol/organelles would likely do the trick). Also, this discussion should be added to the discussion section for a more biochemical / functional accent. On the side: In this aspect it would be -for later studies- interesting to search for the TML/HTML antiporter. This actually was done in our laboratory but never reported/followed up. We identified Slc25a45 as the TML/HTML antiporter. Free information, do with it what you want.

The reviewer is right that proteins catalyzing the movement of molecules across membranes are considered enzymes by the Enzyme Commission (EC 7). However, in this study we focused on the enzymes of the biosynthetic pathway. In the revised version we mention that carnitine biosynthesis in mammals is conducted partly in the mitochondria and partly in the cytosol and the possible existence of a TML/HTML antiporter. The determination of the subcellular localization of enzymatic activities would indeed require the fractionation of liver (and/or kidney) homogenates. However, such experiments would be complicated by the ethics committee approval and would not provide evidence on the specific protein catalyzing the enzymatic reaction, which is the focus of our study.

Page 13. According to the subcellular localization of enzymatic activities, carnitine biosynthesis occurs initially in the mitochondria and subsequently in the cytosol. The molecular identity of membrane translocators, responsible for the movements of pathway intermediates between cellular compartments, and particularly of the postulated mitochondrial TML/HTML antiporter ⁶⁶, remains unknown.

Minor points

Page 8, third paragraph: the unit for the Km is somewhat cryptic for SHMT1, please add this to the main text to be able to compare Km in mM or uM, that is what enzymologists like. Looking further, I found the Km of SHMT1 in Supplementary table 4 and found it to be 3,8 mM! That is a lot! [now I understand why you did not mention it :)] Especially when compared to tha1 Km and Kcat. The intracellular concentration of TML and HTML are poorly known but this further suggests that the conversion of HTML into TMABA is a

slow one in humans, in contrast to that in rodents. This could be included in the discussion.

The K_m of SHMT1 (3.8 mM) for HTML was reported in comparison to SHMT2 (0.80 mM) in the main text at page 8. In the discussion of the revised version we mention the low efficiency of these enzymes in the HTMLA reaction. However, we were not surprised by the high K_m value for the cleavage of HTML catalyzed by SHMT1, since this is comparable to the K_m value (4.8 mM) for L-serine, that is the main substrate, obtained in the absence of tetrahydrofolate in the hydroxymethyltransferase reaction catalyzed by the same enzyme (Tramonti et al., 2018 *Biochemistry* 57(51):6984-6996).

Page 12: [...rodents possess a more efficient HTMLA enzyme than humans and other primates] in consideration of the low k_{cat} and high K_m of SHMT for the HTML substrate

Page 12, last paragraph: 75% dietary carnitine is for omnivores, not for vegetarians and especially vegans, where more than 75% comes from biosynthesis (and efficient reabsorption in the kidney). Add omnivores and consider adding this nuance.

We added this specification to the text along with a reference to a recent review underscoring the different apport of carnitine by omnivorous and vegan diets.

Page 13: [... with about 75% of total body carnitine originating from food sources], at least in the presence of an omnivorous diet^{67,68}

Supplementary figure 4: very nice figure. Could this be transferred to the main text? It would be a pity if this would be hidden in the supplemental part, it is a great overview of carnitine biosynthesis distribution in nature. One more question; why is *Rattus Norvegicus* not included? Could this be added if data is available as this is one of the best studied organisms when it comes to carnitine biosynthesis.

We thank the referee for the appreciation of this figure and the proposal to transfer it in the main manuscript. However, in view of the recommendation of other reviewers to put more emphasis in the computer method and less in carnitine biosynthesis, we decided to maintain it as a supplementary figure. As suggested by the reviewer, ***Rattus norvegicus*** has been added to the species analyzed in Supplementary Fig. 4 (5 in the revised version).

Supplementary figure 14: please rearrange the panels and cluster them enzyme- or substrate-wise so that it is easier to understand without the legend. Just add the enzyme name in the graph, please.

We apologize for the untidy figure. The new version of the Supplementary figure 14 (15 in the revised version) has been organized according to the reviewer suggestion.

Reviewer #3 (Remarks to the Author):

This is a manuscript that reports two research stories, one on carnitine biosynthesis in animals that is interesting but more appropriate for a discipline journal, and another on a method to identify function in enzymes, which is innovative and potentially very

impactful. Overall, I think the authors should reframe the manuscript to concentrate on the OSMES approach and how it can be deployed to uncover enzyme function in cells.

In the revised version we tried to convey more effectively the link among the two research stories. We emphasize that not only the method has been instrumental to identify candidate enzymes in the biological case, but also that the biological case has inspired the computer method. In addition, more emphasis to the method and less to the biological pathway is given in the revised version. We point out, however, that the knowledge of how carnitine is made in humans and other animals is important information, given the relevance of this compound in fat metabolism.

Page 11. The design of our structure-based screening was motivated by the existence of orphan reactions in biological pathways in which established bioinformatics methods^{7,11} fail to identify candidate genes, as in the case of the carnitine biosynthesis pathway investigated here.

1) On the generality of the OSMES approach. I agree it has been useful for this example, but the authors need to apply it to different systems to convince the community that this is indeed an approach that has merit. This is particularly important for assessing the best performing metric for ranking particular candidates. The metric here is very specific to PLP enzymes—how might it be applied to other enzyme families?

While OSMES metrics based on binding energy or number of conformations are generally applicable, CFC metrics depend on the catalytic mechanism. According to the reviewer suggestion, we adapted the OSMES procedure to another type of enzymatic reaction (aldehyde dehydrogenase). The results obtained with the validation screening suggest that this approach can be extended to other enzyme families. A new paragraph and supplementary figure (**Supplementary Fig. 21**) and table (**Supplementary Table 10**) have been added to the results section.

Page 11. Extension of the OSMES procedure to other enzymes

To test the possibility of extending the OSMES procedure to a different group of enzymes, we decided to apply OSMES to aldehyde dehydrogenases (EC: 1.2.1.-), a numerous protein family sharing a common catalysis mechanism (Supplementary Fig. 21). A member of aldehyde dehydrogenases, TMABADH, is functionally related to HTMLA, as it catalyzes the subsequent reaction in the biosynthetic pathway (see Fig. 3a). Also in this case we modeled an initial step of the reaction mechanism involving the nucleophilic attack by an active site cysteine to the substrate aldehyde carbon (Supplementary Fig. 21a), with the formation of a covalent intermediate⁶¹. The subsequent transfer of a hydride ion (H⁻) from this thioester intermediate results in the reduction of NAD(P)⁺ to NAD(P)H.

We considered substrate orientations in the active site as CFC when the distance between the aldehyde carbon and the catalytic cysteine thiolate was ≤ 3.5 Å, which is regarded as an upper limit for near attack conformation^{61,62}. Using this condition, we applied OSMES to human and murine aldehyde dehydrogenases encompassing two different PFAM domains (Gp_dh_N: PF00044 and Aldedh: PF00171), with a total of 20 and 22 enzymes for human and mouse, respectively (Supplementary Table 10). Since

in these proteins the active site is enclosed within monomeric units, the oligomerization step was not performed.

Six different aldehyde molecules, which are known substrates of eight different enzymes, were used as positive controls for validation (Supplementary Fig. 21b). We observed a performance similar to PLP-dependent enzymes, with CC-CFC as the best ranking method (AUROC score=0.86) and the energy-based methods (LCE, BCE) as a close second best. Conversely, the ranking methods based on the number of conformations (LCC, BCC) seem to lack discriminative power (Supplementary Fig. 21c, d). In various instances the conformation of the best cluster (BC) corresponded to a catalytically favorable conformation (Supplementary Fig. 21e).

2) In the abstract, as stated the outcome seems to be that a gene has been misannotated—yes this may be true (albeit trivial), but is this really the key impact of the work? Isn't the method by which the real gene is uncovered the key impact?

We agree with the reviewer that the method by which the gene function was uncovered is more relevant than the annotation of the mouse gene. The sentence on the reannotation of the mouse gene has been eliminated from the abstract .

3) The finding that the HTML cleavage is less efficient in humans compared to Tha1 (and the conclusion that humans have lost HTML activity) needs to be carefully considered. Yes the kinetic activity in the in vitro assay appears lower, but a variety of factors in the native cell environment could form a perfectly active enzyme—for example, allosteric activation not accounted for in the assay, gene expression levels, pH in the cell, oligomerization. Similarly, in mice Tha1 may be considerably inactive in the native environment. To demonstrate that humans have actually lost this activity, or that it is knocked down, some kind of comparative in vivo assay needs to be done. Metabolomics may be a way forward. Also, it could be that the authors method of finding the real enzyme in humans has failed.

Our conclusion that humans have lost a gene encoding a protein with efficient HTMLA activity (Tha1), but possess an enzyme (SHMT) able to catalyze the reaction with reduced efficiency, is supported by our in vitro data and by earlier in vivo evidence obtained with the administration of labeled (radioactive) precursors to rodents or humans. Nowadays, in vivo evidence could be obtained with metabolomics, even though with this approach it would be difficult to distinguish among carnitine introduced with the diet or produced endogenously. We agree with the reviewer that in vitro assays do not take cell complexity into account, but it is also very difficult to perform an in vivo assay to measure SHMT activity with HTML under conditions identical to those that exist inside the cell. However, in the revised manuscript we have also assayed SHMT activity with HTML in the presence of tetrahydrofolate (**Supplementary Fig. 16**), to account for the fact that this SHMT cofactor is also present in the cell. Results show that tetrahydrofolate does not affect the cleavage reactions of L-allo-threonine and HTML.

Supplementary Fig. 16: Aldolase activity of human SHMTs in the presence of tetrahydrofolate. Barplot of the aldolase activity of SHMT1 (left) and SHMT2 (right)

towards HTML and L-*allo*-threonine in the absence (white) or in the presence (red) of 40 μ M of tetrahydrofolate (THF).

4) The authors might consider how protein dynamics play a role in enzyme mechanism and how this knowledge could be reflected in their OSMES approach.

Protein dynamics is an important aspect of catalysis. Our approach accounts for some level of conformational changes during the binding by allowing flexibility in key catalytic residues (Methods). An entirely flexible binding site would increase the degree of freedom of the system to a point that an exhaustive sampling of the conformational space during the docking would be impossible. Furthermore, a full consideration of protein conformational changes by means of molecular dynamics simulations would be computationally overly demanding in the screening of several proteins.

5) The Discussion focusses on the carnitine biosynthesis, but this is a less interesting. More consideration should be placed on the OSMES approach and its potential impact. The discussion has been extensively revised to put more emphasis on the limits and potential impact of the OSMES approach. We have, however, maintained a discussion on carnitine biosynthesis, as we consider our results on this aspect of high biological relevance.

Minor comments.

1) The abstract is perhaps too detailed and not accessible to the general reader. I appreciate the tight word limit, but it isn't immediately obvious to a general reader what has been accomplished.

We tried to make the abstract more accessible to the general reader while adhering to the strict word limit.

2) Sentence "This allows to predict..." needs editing.

We have corrected the sentence into "This makes it possible to predict".

3) Are the active site residues formed by both monomers in the dimer? If so, comment on this and the impact/importance of the 'oligomerization' step in the OSMES workflow. According to the reviewer's suggestion, in the revised version we underscore the importance of the oligomerization step, as most of the enzymes in our set have their active site at the dimer interface.

Page 4: [Since most PLP-dependent enzymes belong to fold-type I, which is characterized by obligate dimeric association forming two identical active sites at the interface], oligomerization of the monomeric AlphaFold models is a crucial step of the OSMES pipeline.

4) Introduction. The statement that enzymes bind their substrates with high affinity is not true. This would be anticatalytic. Instead, they bind substrates with sufficient affinity.

Perhaps the authors are referring to specificity, but even then, many enzymes are not particularly specific. Perhaps rejjig this sentence.

We thank the reviewer for this correction. The sentence has been modified in the revised version.

Page 2: Enzymes must bind their substrate molecules with adequate affinity.

5) The SEC-SAXS data is very nice. The SEC trace appears to be asymmetric – is the oligomeric state of the enzyme changing as a function of protein concentration? If so, what is the likely oligomeric state of the protein in the assay (low [protein]).

In the revised version we included new supporting data in **Supplementary Fig. 19** adding evidence that the Tha1/HTMLA protein exists as tetramer in near-native solution condition. The panel **a** of the Supplementary Fig. 19 shows the chromatographic profile of the protein mainly eluting as a monodispersed peak composed of highly pure protein that was subsequently used for crystallization experiments. The same batch was used for the SEC-SAXS experiment at bio-SAXS beamline BM29 at ESRF. The panel **b** shows the elution traces of the protein monitored following the UV at 280 nm and the total scattering. An estimation of the total protein concentration under the peak area indicates an overall concentration of 0.12 mg/mL (corresponding to 3.1 μ M) similar to the concentration used in our enzymatic assays. Even at this low protein concentration it was possible to extract SAXS structural parameters including the radius of gyration, R_g , and a suitable SAXS curve (panel **c**) obtained through averaging of buffer-background subtracted frames across the entire elution traces of the SEC-SAXS experiment. We observe that the eluted particles share a stable R_g and a scattering profile that was fitted with high accuracy using the calculated profile of Tha1/HTMLA tetramer (with a χ^2 of 1.14). Notably, the calculated profile of Tha1/HTMLA dimer displays a χ^2 of 6.74, clearly indicating that the protein exists as tetramer in solution. The *ab initio* shape of the particle in panel **d** shows an occupancy that accommodates well the tetrameric form. The reviewer correctly reported a slight shoulder in the elution peak. This small peak asymmetry can be indicative of different particle-column interaction behaviours occurring at extremely low protein concentrations (i.e. at sub micromolar concentration). We believe this is not due heterogeneity of the sample since we added clear evidence that at the protein concentration conditions similar to those used in the enzymatic assay the protein exists as a tetramer as documented by SEC-SAXS analysis.

Reviewer #4 (Remarks to the Author):

In this work, Malatesta and coworkers develop a computational protocol for identifying enzymes capable of accepting a given substrate and catalyze a particular PLP-dependent reaction. The procedure (OSMES: one substrate-many enzymes screening) is applied to identify 3-hydroxy-N-trimethyllysine aldolase (HTMLA) candidates in human and mice. A few of the identified enzymes are then experimentally tested. In addition to that, a crystal structure is reported for Tha1, which appears to be the most active one. This experimental validation shows that the predicted OSMES ranking is not very accurate, the first two hits for Homo sapiens have very low activity as

shown with the kinetic characterization, and similarly for *Mus musculus*. Although the paper is interesting, in my opinion the protocol is far from being optimal as it does not provide an accurate ranking of the activities, and it is also not clear to me how general the approach is, especially if non-cofactor dependent enzymes are pursued.

As in our approach we considered only a particular aspect of the catalytic process, we agree with the referee that our procedure does not provide an accurate ranking of catalytic activities. A note of caution has been added in discussion. However, the purpose of our screening was rather the identification of suitable candidates for a specific enzymatic reaction on a structural basis. In this respect, we consider the first two hits in *Homo sapiens* a good result, as these are to our knowledge the only two human proteins able to catalyze the investigated reaction. Similarly, we consider a good result the ranking obtained in *Mus musculus*, especially when the Tha1 crystal structure is taken into account. We also agree with the referee that our approach is directed towards a particular group of enzymes that use the PLP cofactor. However, the results obtained with aldehyde dehydrogenases included in the revised version suggest that a similar approach can be extended to different groups of enzymes with known catalytic mechanisms [see also our response to reviewer #3].

Page 11. Since our approach takes into account a single aspect of the catalytic cycle -the formation of a catalytically competent enzyme-substrate complex- it is not anticipated to provide an accurate ranking of enzymatic activities. We have, however, observed that it can aid in the identification of enzymes capable of catalyzing a specific reaction by ranking them in the top positions within a set of proteins.

Page 12. Generally, the determination of catalytically favorable conformations relies on prior knowledge of the catalytic mechanism, which is accessible for a subset of evolutionarily distinct enzymes⁵⁸. The suitability of a CFC method in the screening of enzymatic reactions should thus be assessed on a case-by-case basis.

I have the following comments:

1. In the OSMES protocol, they use AF2 to generate the 3D structures, but then use SWISS-MODEL oligomerization states for constructing the oligomers. Why not using AF2 multimer from the first place?

When this study was initiated AF Multimer was not available. In the revised manuscript, we analyzed with OSMES the enzyme set for both organisms obtained with AlphaFold Multimer. The results, although similar, are slightly worse than those obtained with superposition of Swiss-Model oligomers (AUROC=0.82 Vs. 0.84). [See also our response to reviewers #1]

Page 5: As an alternative for the second step of the OSMES pipeline (the assembly of oligomeric structures), we considered the use of AlphaFold Multimer³⁶. Also in this case, CC-CFC was the best ranking method. However, a slight decrease in the performance was observed with respect to the use of oligomers based on SWISS-MODEL templates (AUROC=0.82, Supplementary Fig. 4).

2. The protocol constructs a box around the catalytic lysine for docking the PLP-external aldimine. However, many of the X-ray structures of PLP-dependent enzymes are crystallized in the internal aldimine form (Lys covalently bound to PLP), which means that the conformation of the lysine might be different from the one it adopts in the external aldimine intermediate. In addition to that, the external aldimine is optimized using a forcefield instead of DFT (or semi empirics). How good/reliable are the docked external aldimine structures? The comparison of a predicted pose with an X-ray structure with an external aldimine (or similar) would reveal interesting insights in this regard. In Fig. S2 some structures for the validation PLP-external aldimine are shown, but it is hard to assess the quality of these structures. The same happens with the PLP-external aldimine docked in the active site pocket that are shown in figures S7 and S8. The authors should provide more quantitative descriptors (distances, angles, RMSD, etc). **AlphaFold models were constructed without assuming the presence of PLP. To account for the different conformation of the catalytic lysine in the presence of the external aldimine, the catalytic lysine was maintained flexible in the docking. Supplementary Fig. 2 is meant to illustrate the library of substrate-PLP used in the validation screening. These molecules were minimized using a forcefield. However, the docked conformations were further optimized by Autodock allowing free rotation of the rotatable bonds. In Fig. S7 (now Fig S8) we aimed to demonstrate that the position of the PLP cofactor in the docking poses generally aligns with crystallographic structures. Unfortunately, most structures involve the internal aldimine, except for SHMT2 in complex with GLY-PLP (8AQL), where a good agreement with the docked pose was observed (RMSD=0.90 Å). For a more quantitative description, RMSD values for the PLP atoms are provided for all comparisons in the figure.**

Supplementary Fig. 8, legend:

**[...] The RMSD obtained by the comparison of the PLP atoms are the following:
SHMT2: 0.90 Å, Tha1: 1.39 Å, SHMT1: 0.74 Å, SGPL1: 2.09 Å, KYAT3: 1.09 Å KYAT1:
1.64 Å.**

3. They have experimentally tested the most efficient candidates from Homo sapiens and Mus musculus and found that the candidate in mice is the most efficient for the reactivity they are seeking, while the top two candidates from Homo sapiens are less efficient. Based on this, they conclude that humans have lost the gene involved in Carnitine synthesis. Is this stating entirely accurate? It could be that with the method they have used, they haven't been able to find the best candidate, but a better one to the ones identified could exist.

As the full set of PLP-dependent enzymes was considered in our analysis and our results converge with previous evidence in carnitine biosynthesis, the existence of a better HTMLA candidate overlooked by our analysis is deemed possible, but not very likely.

4. As mentioned before, OSMES ranking is not very accurate, as the most efficient enzyme for mice (called Tha1) is ranked 9th. However, once they obtain the X-ray structure and use it in the OSMES method, they find that Tha1 is ranked second. What

differs between the model and X-ray structure? This might suggest that the used models are not of high accuracy.

Fig. 6c has been modified to emphasize the difference of active site residues between the model and X-ray structure. Data reported in Supplementary figure 18 suggest that the models used in the screening have different levels of accuracy, with Tha1 being among the worst ones.

Fig. 6c, legend:

[...] Residues showing a different position in the model and crystallographic structure are labelled. [...]

5. It is not clear to me why the better ranked enzymes in the case of mice were not experimentally characterized. In addition to that, the experimental characterization of enzymes ranked at the low and mid positions in the ranking might also provide relevant information on the quality of the ranking developed.

We did not characterize the high-ranked mouse Shmt1 and Shmt2 assuming that their activity is equivalent to the human orthologs, also supported by the similarity of active site residues (Supplementary Figure 7). In the revised version, additional genes were considered for the activity assays. ABAT as an example of high-ranking hit (4th in mouse and 3rd in human), Thnsl2 and Oat as examples of mid-ranking hits, and PSAT1 as an example of low-ranked hit. We obtained a soluble expression only with the three latter proteins. The additional results are shown in **Fig. 5c, Supplementary Fig. 11g-i** and described in the **“Biochemical validation of HTML-OSMES candidates”** section of Results.

Page 8: In addition, we considered screening candidates without previous evidence of aldolase or beta-lyase activity: human ABAT as an example of high-ranking hit, mouse Thnls2 and Oat, as mid-ranking hits, and human PSAT1 as a low-ranking hit. [...] We obtained soluble expression for all the proteins with the exception of ABAT.

6. In the validation step, the authors find that within, the CC-CFC distribution aminotransferases achieved the worst results as compared to other reactions. Do the authors have an explanation for that?

Aminotransferases are generally promiscuous enzymes capable of catalyzing the same reaction with different substrates. In addition, their reversible mechanism also makes their active site well accessible to products, which is not the case for other enzymes (decarboxylase, aldolase, etc..) This may explain why the ranking of aminotransferases is less accurate than for other reactions.

Minor notes:

1. It is stated that some bonds of the PLP-HTML complex were allowed to rotate (Fig. 3b). However, I could not find the information on the rotatable bonds in Fig. 3b.

We have modified Figure 3b, coloring the 13 rotatable bonds in green and updated the legend.

REVIEWERS' COMMENTS

Reviewer #1 (Remarks to the Author):

In this revised manuscript, the authors adequately addressed this reviewer's initial concerns. However, there are a few minor comments arising from the new version and added experiments.

-Fig 2E: "black dashed line delimits the top 10 positions" but is not immediately visible on the graph

-Sup Fig 11i: PSAT1 does not have a discernable peak in the 400-430nm range, but rather is shifted downfield near 340nm. If there is a peak, a zoomed-in view would be helpful. If not, did the authors confirm this protein is active in general?

-Page 10, line 407: "The major structural difference is related to an insertion of 13 residues in Tha1 between positions 346-260"...these numbers appear to be misannotated

Reviewer #2 (Remarks to the Author):

My comments were addressed appropriately.

Reviewer #3 (Remarks to the Author):

The authors have addressed my concerns.

Reviewer #4 (Remarks to the Author):

In this new revised version of the manuscript the authors have further demonstrated the general applicability of the OSMES approach by applying the procedure to a different group of enzymes, aldehyde dehydrogenases. They observed a similar trend as in the case of PLP-dependent enzymes and concluded that CC-CFC is the best ranking method. They have also provided further evidence for the quality of the predicted poses versus crystallographic structures, as well as more information on the AF2 models and accuracies. As proposed, some additional genes were considered for the activity assays including mid-ranked hits, and a low-ranked one. Although I believe that if the method could provide an accurate ranking of enzymatic activities would be substantially more impactful, I agree with the authors that the developed protocol can aid in identifying enzyme hits for catalyzing a specific reaction of interest, especially if one then experimentally tests some of the better ranked identified hits. I believe this new revised version of the manuscript can now be published in Nat. Commun. without any further revision.

We thank the reviewers for the positive evaluation of our manuscript. Our responses to the Reviewer #1 remarks are in blue.

REVIEWERS' COMMENTS

Reviewer #1 (Remarks to the Author):

In this revised manuscript, the authors adequately addressed this reviewer's initial concerns. However, there are a few minor comments arising from the new version and added experiments.

-Fig 2E: "black dashed line delimits the top 10 positions" but is not immediately visible on the graph

The missing dashed line has been added as described in figure legend.

-Sup Fig 11i: PSAT1 does not have a discernable peak in the 400-430nm range, but rather is shifted downfield near 340nm. If there is a peak, a zoomed-in view would be helpful. If not, did the authors confirm this protein is active in general?

A panel with a zoomed view showing evidence of the peak in the 400-430 nm range has been added to Sup. Fig. 11.

-Page 10, line 407: "The major structural difference is related to an insertion of 13 residues in Tha1 between positions 346-260"...these numbers appear to be misannotated

The misannotated positions have been corrected.

Reviewer #2 (Remarks to the Author):

My comments were addressed appropriately.

Reviewer #3 (Remarks to the Author):

The authors have addressed my concerns.

Reviewer #4 (Remarks to the Author):

In this new revised version of the manuscript the authors have further demonstrated the general applicability of the OSMES approach by applying the procedure to a different group of enzymes, aldehyde dehydrogenases. They observed a similar trend as in the case of PLP-dependent enzymes and concluded that CC-CFC is the best ranking method. They have also provided further evidence for the quality of the predicted poses

versus crystallographic structures, as well as more information on the AF2 models and accuracies. As proposed, some additional genes were considered for the activity assays including mid-ranked hits, and a low-ranked one. Although I believe that if the method could provide an accurate ranking of enzymatic activities would be substantially more impactful, I agree with the authors that the developed protocol can aid in identifying enzyme hits for catalyzing a specific reaction of interest, especially if one then experimentally tests some of the better ranked identified hits. I believe this new revised version of the manuscript can now be published in Nat. Commun. without any further revision.